# Study of the Mineralogical and Chemical Compositions of the Weakly Magnetic Fractions of the Egyptian Black Sand Altered Ilmenite

**Mohamed Ismail Moustafa** [1,2]

1    Department of Civil Engineering, College of Engineering, Northern Border University,
     Arar 91431, Saudi Arabia; ismail2251962@yahoo.com
2    Physical Concentration Department, Nuclear Material Authority, El Maadi, Cairo P.O. Box 530, Egypt

**Abstract:** One of the most extensively studied topics in dozens of studies is the alteration process of ilmenite, the formation of leucoxene, and the presence of some impurity oxides: $SiO_2$ and $Al_2O_3$. The altered Egyptian black sand ilmenite grains of relatively lower magnetic characters are studied using the binocular microscope and the Cameca SX-100 microprobe instrument. Both individual brown- and black-altered grains separated at 0.5 and 1 ampere values are investigated. The detection of the various alteration phases, their molecular formulas and limits, and the role of $SiO_2$ and $Al_2O_3$ in alteration mechanisms are detected. The alteration phases include pseudorutile (psr) and leached pseudorutile (lpsr) of different phases in addition to rutile. Few analyzed spots are detected to be leached ilmenite (lilm). Several Excel software are adopted to calculate the chemical formulas of each alteration phase. The contents of $TiO_2$ and $Fe_2O_3$ of all the investigated psr/lpsr in the study are in the range of 59.16–86.56% and 37.3–6.68%, respectively. The Ti/(Ti + Fe) ratio for these formulas ranges between 0.60 and 0.88. The psr/lpsr chemical formulas of all the investigated grains range as follows: $Fe_{2.01-0.50}Ti_3O_{8.97-4.50}(OH)_{0.03-4.50}$. The concluded lowest cationic iron content of the well-defined accepted lpsr phase is 0.5 with a corresponding molecular formula of $Fe_{0.50}Ti_3O_{4.5}(OH)_{4.5}$. The results revealed that in the region of 68–70 $TiO_2$%, the mechanism of ilmenite alteration may be changed where neither all the analyzed $TiO_2$ of the spot nor all the calculated structural water are contained within the molecular formula of lpsr. There are other associated mineral phases containing some $TiO_2$ and also some structural water which most probably are removed from the lpsr phase. As the analyzed spots are located at highly fissured locations, the alteration process is relatively faster and the lpsr phase can be broken into rutile and hematite. Additionally, as the analyzed $TiO_2$ and structural and/or molecular water contents increase, the darkness of the BSE image areas of the grain increases; this may reflect the existence of an individual $TiO_2$ phase, most probably rutile, mixed in homogeneity with the existed lpsr component. As the content of $TiO_2$ increases, within a definite $TiO_2$ range (80–85%), the associated contents of $Al_2O_3$ and $SiO_2$ increase. When the contents of the structural and/or molecular water contained within the lpsr phases decreases, the total oxides sum is more than 98%, the contents of $Al_2O_3$ are highly depleted. In the late alteration stages, the lpsr structure does not suddenly collapse but gradually produces other associated mineral phases. The relatively enriched contents of $SiO_2$ and/or $Al_2O_3$ in some secondary rutile grains can be explained as most of the $SiO_2$% is associated with mol water or bearing for mol and/or str water necessary for the leachability of $Fe^{3+}$ from the psr structure. The XRD patterns of the investigated grains before and after roasting at 1100 °C for one hour are detected and interpreted.

**Keywords:** altered ilmenite; leached ilmenite; leucoxene; pseudorutile; leached pseudorutile; exsolved intergrowth; X-ray diffraction

## 1. Introduction

The most abundant economic mineral of the Egyptian black sand is ilmenite followed by magnetite, garnet, zircon, rutile, and monazite in decreasing order of abundance. The

Egyptian beach ilmenite partly altered to a greyish material which in a later stage is replaced by a very fine intergrowth of rutile–anatase product with a microporous structure [1]. Two post-depositional alteration varieties of the Egyptian beach ilmenite can be distinguished. The first is the amorphous iron–titanium oxides due to alteration along borders and cracks, or as thin rim separating fresh ilmenite relics from a broad outer zone of leucoxene. The second variety is the alteration to leucoxene; a very fine crystalline aggregate with a mottled appearance [2]. Highly leucoxenated secondary rutile grains within the obtained rutile concentrate, as an end product of ilmenite alteration, were identified by [3,4]. Several types of magnetic and non-magnetic altered ilmenite varieties were detected by [5]; the author provided an explanation for the formation of each of these altered varieties.

Many papers were introduced about the alteration products of ilmenite. The discovery of a new mineral, arizonite, with a chemical composition closely corresponding to $Fe_2O_{3.3}TiO_2$, was reported by [6]. It was considered as merely weathered ilmenite [7]. The nature and chemical composition of leucoxene are known to be variable [8–18]. The algorithm for calculating formula units in the mineral series of leached ilmenite–pseudorutile and leached pseudorutile was given by [19].

The name pseudorutile was proposed for the product of oxidation and the progressive partial removal of iron due to alteration of ilmenite giving an intermediate iron titanate of a definite structure [20].

The ilmenite concentrates from several beach sand deposits were studied and it was concluded that most concentrates contain ore grains ranging from fresh ilmenite to a highly altered product approaching pure $TiO_2$ in composition [21].

The abundance co-occurrence of pseudobrookite and altered ilmenite was reported in Quilon sands, India [22]. However, [23] does not agree with the explanation of [21], unless the primary pseudobrookite was present early on in the source area.

In the case of an ilmenite alteration with $Ti/(Ti + Fe) > 0.7$, the Al and Si levels increase rapidly with increasing $Ti/(Ti + Fe)$ ratios, to maximum values near 1.5 wt% $Al_2O_3$ and 0.5 wt% $SiO_2$. This increase is due to co-precipitation or adsorption of these elements from the surrounding soil solutions onto the freshly formed alteration products [24].

The alteration of ilmenite to psr decreases the cell volume by up to 13% [25], while it is reported to be 6% by [26], and reaches 40% by alteration to leucoxene leading to shrinkage cracks. The XRD pattern of psr in comparing with rutile is similar with two additional lines of d-values 2.71 Å and 2.98 Å [24]. There is a more extensive network of nanopores in the hydroxylian pseudorutile (HPR) than those of psr [27].

In studying a sample from a Rosetta ilmenite concentrate of Egyptian black sand, it was explained that although the alteration to psr is observed, further alteration to leucoxene is very rare [28]. Most of the mineralogical features for both of the homogeneous Egyptian black sand ilmenite, the different exsolved intergrowths between ilmenite-other mineral components, and the partially altered ilmenite were explained [29,30]. In these two last studies, it was noticed that the presence of molecular water is very important in the alteration process of ilmenite or some associated silicate mineral impurities. Additionally, the chemical composition of the highly altered leucoxenated Egyptian beach ilmenite grains reflects that the $TiO_2$ ranges between 59.45 and 89.72%, the total iron content ($Fe_2O_3$) varies from 2.34 to 32.68%, whereas the $SiO_2$ content varies from 0.89 to 8.19% [31]. In the present article, some of the weakly magnetic altered ilmenite grains are investigated to explain their mineralogical and chemical composition characters. The purpose of the work is to detect both of the different lpsr phases, the lowest iron content in these phases, and the most stable molecular formula of the detected lowest lpsr phase, as well as the prediction of the role of $SiO_2$ and $Al_2O_3$ contents in ilmenite alteration. Are they just impurities as reported in all previous studies or do they play a definite role in ilmenite alteration stages. In fact, if these targets are explained, then both of the most accepted formed lpsr molecular formulas, real role of $SiO_2$ and $Al_2O_3$ contents in mineral alteration, and illustration of some given previous psr/lpsr molecular formulas can be achieved. Several

Excel software are constructed for concluding the chemical formulas of the various ilmenite alteration products.

## 2. Materials and Methods

A large bulk sample of the surficial naturally highly concentrated beach black sand was collected from the beach area at the Mediterranean coast, 7 km to the east of the Rosetta estuary, Egypt. Using the difference in physical characters between the various economic minerals [5,29,30,32,33], the collected surficial naturally highly concentrated beach raw sands were processed using the following equipment:

- The Reading cross-belt magnetic separator for magnetic separation where the raw sample is differentiated into three fractions: a ferromagnetic fraction, a bulk magnetic fraction, and a bulk non-magnetic fraction.
- The full-size Wilfley shaking tables for wet-gravity concentration of the obtained bulk non-magnetic fraction.
- The Carpco (HP 167) high-tension roll-type electrostatic separator for treating the obtained tabled concentrate to obtain a bulk conductor rutile fraction and a bulk non-conductor zircon fraction.
- The Carpco (MIH 13-231-100) industrial high-intensity induced roll dry magnetic separator for magnetic separation of the obtained bulk rutile conductor fraction.
- The Frantz isodynamic magnetic separator; the obtained three successive magnetic fractions of the last treatment stage were mixed together as a bulk magnetic fraction. It is composed of hematite, ilmeno–hematite, different varieties of magnetic primary rutile [33], various grades of altered ilmenite grains, in addition to a minor amount of Cr-bearing minerals and other magnetic minerals [5]. A relatively smaller representative sample was obtained from the bulk magnetic fraction and subjected to magnetic differentiation using the Frantz isodynamic magnetic separator where the used adjustment of operating conditions was a longitudinal slope of $20°$, side slope of $5°$, feeding rate of 30 g/h, and different successive ampere values of 0.1, 0.2, 0.25. 0.35, 0.5, and 1, where six magnetic fractions and only one non-magnetic fraction were obtained. The altered ilmenite grains obtained in the two individual magnetic fractions separated at 0.5 and 1 ampere values were investigated and included in this article.
- The microscopic investigation. The altered ilmenite varieties obtained for these separated two individual magnetic fractions were investigated using the binocular and reflected microscopes.
- The microprobe analysis. The investigation of the different altered ilmenite grains was carried out by a Cameca SX-100 electron microprobe analyzer (EMPA), Institute of Mineralogy and Crystal Chemistry, Stuttgart University, Germany. The microprobe instrument is equipped with three wavelength dispersive spectrometers (WDS) and an energy dispersive spectrometer (EDS). The whole surface of the polished sections was examined by backscattered electron (BSE) images, so that grain with, for example, a 10 μm size or even smaller, could be detected. The analytical conditions were 15 kV accelerating voltage; 15 nA electron current; a 180s counting time for each analyzed spot in the investigated grains and a focused electron beam diameter of 1 to 4 μm. The following standards were used: diopside for Mg and Ca, albite for Na, corundum for Al, orthoclase for Si and K, rutile for Ti, rhodonite for Mn, $Fe_2O_3$ for Fe, $Cr_2O_3$ for Cr, V for V, and sphalerite for Zn. Lines used for analysis were Kα for each of the analyzed elements. For each detected altered ilmenite variety, a definite number of grains were picked individually and polished for the investigation using the microprobe.
- The dependance of chemical analyses to predict mineral composition and/or conclude a molecular formula for a definite mineral phase has been followed by many authors. Taking a definite mineral phase into consideration, there is not a definite type and number of the required analyzed oxides except that all the major oxides composing the mineral phase must be included. In the case of detecting the ilmenite alteration, the

electron microprobe analyzer (EMPA) was used for a definite adjustment of operating conditions. Most of the analyzed oxides in the present study, except for $Na_2O$ and $K_2O$, were included in two previous studies [29,30]. The study of [25] used the elemental composition by means of microprobe analysis and the calculation for the psr and lpsr formula on the basis of 3 Ti. They act with similar analyzed oxides of the present study except ZnO and $K_2O$. However, NiO was included in some of their analyses. On the other hand [26], acting with 9 analyzed oxides, where both of $P_2O_5$ and $ThO_2$ were among them, they are not included in the present study. $ZrO_2$, $V_2O_5$, and $P_2O_5$ are included with the analyzed oxides of [24]. Only 7 analyzed oxides, namely $TiO_2$, $Fe_2O_3$, MnO, MgO, $SiO_2$, CaO, and $Al_2O_3$ were used to study the grains of hydroxylian psr [34]. On the other hand, 14 analyzed oxides were included in the studies of [19,35]; 5 out of them are not included in the present study which are NiO, $Nb_2O_5$, $Ta_2O_5$, $V_2O_3$, and SnO. However, both $Na_2O$ and $K_2O$ are not included with them [19,35].

- The X-ray diffraction instrument (XRD). The Philips X-ray generator (PW 3710/31) with automatic sample changer (PW 1775; 21 position) using a scintillation counter, Cu-target tube, and Ni filter at 40 kV and 30 mA was used. This instrument is connected to a computer system using an X-40 diffraction program and ASTM cards for mineral identification.

However, all the used constructed Excel software for the calculation for different molecular formulas of mineral phases, which is the final result of the essence and algorithm of calculations, are given in the article as Supplementary Materials (Files S1–S6). These constructed Excel software includes the calculation for the molecular formula of ilmenite, leached ilmenite, pseudorutile, and leached pseudorutile. In pseudorutile and leached pseudorutile Excel software, the calculation for the number of oxygen anions, number of hydroxyl anions, the content of OH wt%, and hence, the corresponding content of $H_2O$ wt%, are explained.

## 3. Results and Discussion

### 3.1. The Separated Magnetic Fraction at 0.5 Ampere

The magnetic fraction is composed mainly of the black- and brown-coloured altered ilmenite varieties. The black grains are more abundant, angular, relatively finer, and with highly pitted surfaces compared to the brown-coloured grains. The grains of light brown, reddish, and yellowish brown colours are highly increased in the fraction [32]. A considerable number of the black-coloured grains are stained to be partially coated; from 5 to 90% of the grain's surface, with relatively lighter soft-coloured material (dark brown, yellowish, and reddish brown). Other grains are stained or coated with silica.

Taking the brown-coloured grains of the magnetic fraction into consideration, three groups of grains are identified in the following sections.

#### 3.1.1. The Detected Dark Brown Grains at 0.5 Ampere

Seven dark brown grains were investigated. Some of the detected spots are either altered silicate minerals, or individual phases of $TiO_2$ and $Fe_2O_3$ obtained due to the breakdown of an existing lpsr.

The psr and lpsr spots of the grains (Figure 1a–c and Table 1) have $TiO_2$ contents ranging between 68.61 and 84.11% while $Fe_2O_3$ contents range between 6.68 and 20.7%. Their cationic iron ranges between 1.07 and 0.42 while the Ti/(Ti + Fe) ratio ranges between 0.74 and 0.88. The analyzed spot 1 of grain 1c is an impurity of a silicate mineral. All the analyzed spots of the grain (Figure 1a) are lpsr. In grain 1b, the $TiO_2$ content attains 79.01% while the $Fe_2O_3$ content decreases to 12.72%. In grain 1c, the $TiO_2$ content attains 84.11%, while the $Fe_2O_3$ content decreases to 6.68%. The cationic iron content ranges between 0.42 and 0.49. In each analyzed spot of the two grains (Figure 1b,c), the new total sum of oxides (NT), after applying the constructed psr Excel program, is greater than 100%: 104.5–106.11% for grain 1b and 106.8–108.3% for grain 1c. Then, either the calculated structural water or the content of the most abundant analyzed oxide ($TiO_2$) as included

completely in the contained psr phase is incorrect. In both grains (Figure 1b,c), comparing the original total sum of oxides (OT), before applying the constructed psr Excel program and NT for each of them, it is difficult to accept the calculated values for the corresponding contained str water (Table 1). The lowest OT value is 93.8% which reflects a maximum value of 6.2% for str and/or mol water. On the other hand, the highest calculated $H_2O$% by using the adopted psr Excel program is 13.73%. It is noticed that in the lpsr structure, on reaching a $TiO_2$ content around 68–70%, the mechanism of ilmenite alteration seems to be changed. Moreover, as the analyzed $TiO_2$ content increases, the darkness of the BSE image increases from 1a to 1c. This may reflect the existence of an individual $TiO_2$ phase, most probably rutile, mixed with the survived lpsr component. Furthermore, as the content of $TiO_2$ content increases (within a definite range of 80–85%), the contents of $Al_2O_3$ and $SiO_2$ increase.

The careful investigation of the three grains reflects that several spots of them are composed of individual phases for $TiO_2$ and $Fe_2O_3$. Thus, not all the analyzed $TiO_2$ and/or $Fe_2O_3$ are included in psr phases only.

In spots 3 and 4 of the grain (Figure 1d, Table 1), the change in alteration mechanism is very obvious at $TiO_2$ contents in the range of 69–70%. In each of 1 and 2, the content of $TiO_2$ is very close to the mixed contents of $Fe_2O_3 + SiO_2 + Al_2O_3$. However, the investigation of the BSE images for the two spots reflects the breakdown of lpsr into two individual phases of $TiO_2$ and $Fe_2O_3$ which may be mixed with $SiO_2$ and $Al_2O_3$. The detected psr and lpsr spots of the grain have the chemical formulas of $Fe_{1.43-0.57}Ti_3O_{7.46-4.73}(OH)_{1.54-4.27}$ (Table 1).

In the grain (Figure 1e), the spots from 1 to 5 correspond to a definite silicate mineral, which contains some amount of $TiO_2$. In these five spots, as the contents of $SiO_2 + Al_2O_3 + K_2O$ decrease, the contents of $TiO_2$ increase (Table 1). Spots 6, 7, and 8 are lpsr. The mechanism of alteration for spot 8 is different than that of the other two spots. In fact, the calculated $H_2O$% seems to be incorrect where the sum weight% of NT equals 107.7%. Each of spot 9, 10, 11, and 12 is an individual $TiO_2$ phase mixed with minor $Fe_2O_3$. It seems that the grain (Figure 1e) was originally composed of major rutile component, minor exsolved intergrowths of ilmenite which altered to lpsr, and inclusions of a definite contained silicate mineral. In comparing the contents of $SiO_2$ and $Al_2O_3$ and the values for OT and NT of the spots from 9 to 12, it is noted that an appreciable content of only $Al_2O_3$ is favorable with $TiO_2$, especially in the presence of str and/or mol water (Table 1). However, the detected psr and lpsr spots of grain 1e have the chemical formulas range of $Fe_{1.58-0.68}Ti_3O_{7.69-5}(OH)_{1.31-4}$ (Table 1).

In the grain (Figure 1f), the analyzed spots from 1 to 6 correspond to a definite silicate mineral containing some content of $TiO_2$. Due to its alteration, considerable contents of $SiO_2$, $Al_2O_3$, CaO, MgO, and $K_2O$ are lost and an enrichment of $TiO_2$ from 6.68 to 72.26% is obtained. It should be noted that if these spots, especially spot 6, are investigated in the absence of their BSE images, they may be identified as lpsr. On the other hand, the spots from 7 to 21 are lpsr while the spots from 22 to 24 are individual $TiO_2$ phases; most probably rutile, after the collapse of an existing lpsr. In these last three spots, the contents of $Al_2O_3$ and $SiO_2$ almost equal zero where the values for OT range between 99 and 100.5%. Then, in comparing them with the spots from 9 to 12 of the grain (Figure 1e), it is observed that with the removal of str and/or mol water, the presence of $Al_2O_3$ and $SiO_2$ with $TiO_2$ (more than 85% $TiO_2$) is not favorable. The detected psr and lpsr spots of the grain have the chemical formulas range of $Fe_{1.45-0.49}Ti_3O_{7.29-4.44}(OH)_{1.71-4.56}$ (Table 1).

In the grain (Figure 1g), the analyzed spots from 1 to 13 are psr and lpsr. The $TiO_2$ contents range between 65.75% and 70.65%, the $Fe_2O_3$ contents range between 29.7 and 21.75%, and the Ti/(Ti + Fe) ratio ranges between 0.68 and 0.74. The contents of $SiO_2$, $Al_2O_3$, MgO, and CaO are relatively lower while the contents of MnO are relatively higher than those of the other analyzed spots from 14 to 20. Additionally, in comparing the values for OT and NT, especially for spots 11, 12, and 13, it can be observed that the calculated structural water contents are not correct and the mechanism of psr alteration is changed. In other words, there are individual phases other than psr. The analyzed spots from 14

to 20 are lpsr. Their $TiO_2$ contents range between 74.54 and 79.66%, $Fe_2O_3$ contents range between 13.55 and 7.09 wt%, and the Ti/(Ti + Fe) ratio ranges between 0.8 and 0.88. The mechanism of alteration for the first six spots is similar to that of spots 11, 12, and 13. Spot 20 has a relatively darker BSE image than others, it is composed of an individual $TiO_2$ phase, most probably rutile.In these seven spots, the contents of $SiO_2$, MgO, MnO, and CaO in addition to $Fe_2O_3$ are at a minimum while the content of $Al_2O_3$ is still relatively higher and the spot still contains structural and/or molecular water; OT equals 96.19%. The spots have the chemical formulas of $Fe_{1.43-0.42}Ti_3O_{7.25-4.26}(OH)_{1.75-4.74}$ (Table 1). However, many of the spots of grain 1gare not totally psr/lpsr phases. Other individual phases of $TiO_2$ and $Fe_2O_3$ are also contained.

In the detected seven grains (Figure 1a–g), the analyzed spots have the chemical formulas in the range of $Fe_{1.58-0.42}Ti_3O_{7.69-4.26}(OH)_{1.31-4.74}$ (Table 1). The Ti/(Ti + Fe) ratio ranges between 0.66 and 0.88.

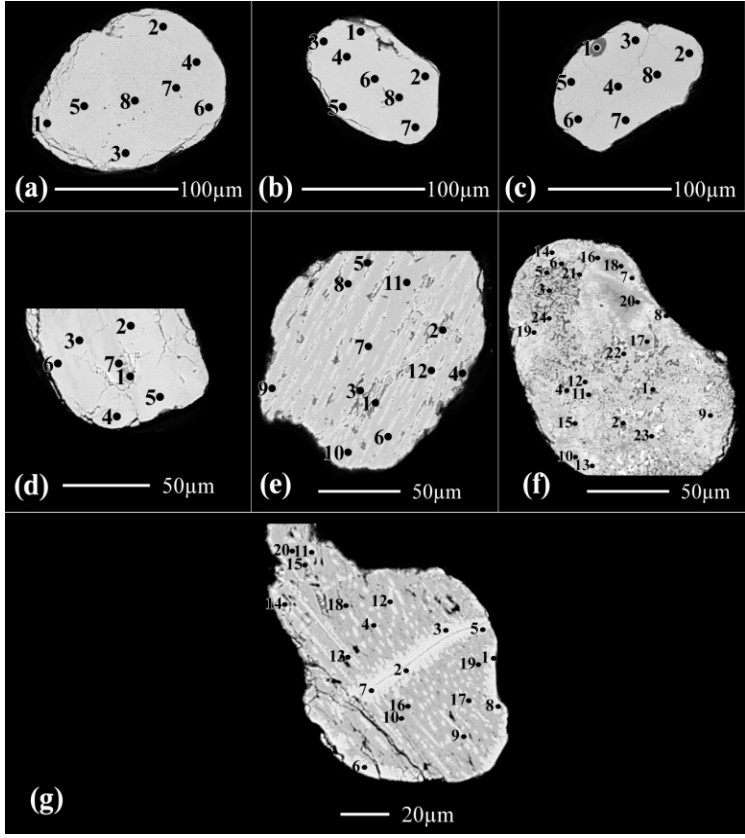

**Figure 1.** The backscattered electron (BSE) images for the altered brown ilmenite grains (**a**–**g**), and the locations of their analyzed spots, separated as magnetic fraction at 0.5 ampere value.

However, according to the BSE images for the grains in Figure 1, the spots with the lowest iron contents (0.49–0.42) are composed mainly of individual $TiO_2$ and $Fe_2O_3$ phases due to a collapse in the majority of contained psr/lpsr phases. Thus, the cationic Fe contents ranging between 0.42 and 0.49 are discarded as minimum iron content values of the still-contained psr/lpsr phases.

3.1.2. The Detected Brown and Yellowish Brown Grains at 0.5 Ampere

Fourteen grains were investigated (Figure 2, Table 1). All the analyzed spots of grain (Figure 2a) are lpsr. The calculated structural water contents of the calculated molecular formulas are not accepted.

In the grain (Figure 2b), spots 1, 2, and from 4 to 7 are lpsr. Spots 3 and 8 are located at highly fissured locations enriched with voids which may accelerate the alteration of lpsr

and its broken into individual $TiO_2$ and $Fe_2O_3$ phases. A considerable amount of $Fe_2O_3$ is lost and/or replaced with $Al_2O_3$. The contents of MnO in spots 3 and 8 are similar for the MnO content of the spots from 9 to 13 indicating that these two last spots may also belpsr collapsed into individual $TiO_2$ and $Fe_2O_3$ phases.

In the grain (Figure 2c), the spots from 1 to 4 are composed of lpsr and have relatively lighter BSE image tints, relatively lower contents of $Al_2O_3$, $SiO_2$, CaO, and $Cr_2O_3$ and relatively higher contents of MnO, and vice versa for the spots from 5 to 8.

The investigated psr/lpsr spots of the three last grains (Figure 2a–c), have contents of $TiO_2$, $Fe_2O_3$, Fe, and Ti/(Ti + Fe) ratio range between 65.07–79.7%, 28.96–13.05%, 1.49–0.7, and 0.67–0.81, respectively. The detected lpsr phases inside them have the chemical formulas in the range of $Fe_{1.49-0.7}Ti_3O_{7.4-5.11}(OH)_{1.6-3.89}$ (Table 1).

The analyzed spot 1 of the grain (Figure 2d) is a definite silicate mineral while spots 2 and 3 are sphene. Both of the two spots 4 and 7 are mainly rutile with minor sphene. Spots 5 and 6 in addition to those from 8 to 12 are rutile with minor amounts of $Fe_2O_3$ and sphene. Investigating the values for OT for these spots reflects the presence of structural and/or molecular water. In addition, considerable contents of $Al_2O_3$, $SiO_2$, and CaO are detected with the relatively higher content of $TiO_2$. Then, both of the preexisting silicate mineral and sphene are altered and replaced with the enriched $TiO_2$ phase, most probably due to definite hydrothermal solutions affecting the source rock bearing area for them.

In the grain (Figure 2e), the analyzed spots 1 and 2 are lpsr but of different alteration phases. The analyzed spot 3 is inside a void while the analyzed spot 4 is beside the edge of the grain where the rate of alteration may be highly accelerated and the lpsr is broken into two individual phases of $Fe_2O_3$ and $TiO_2$. Most of the contained ferric iron phase is leached out. The grain (Figure 2e) seems was originally ferriilmenite–titanhematite exsolved intergrowth where most of the hematite content is leached causing partially empty voids of various sizes and shapes. On the other hand, the ferriilmenite component is altered to lpsr. Comparison between the surrounding environment outside of spot 3 and the other three spots, 1, 2, and 4, may reflect the association of $Al_2O_3$ and $SiO_2$ at the spot location characterized by a relatively faster rate of alteration. In the grain (Figure 2f), the analyzed spots 1, 3, and 4 are lpsr. They have relatively lower contents of $SiO_2$, $Al_2O_3$, and CaO and relatively higher contents of MnO and MgO. Both spots 2 and 5 have almost similar contents of $TiO_2$ such as that of spots 1, 3, and 4. In contrast, the last two spots have relatively lower contents of $Fe_2O_3$, MnO, and MgO and higher contents of $SiO_2$, $Al_2O_3$, and CaO. In addition, the two spots have relatively higher contents of str and /or mol water. The value for OT of spot 2 equals 91.09% and that for spot 5 equals 90.89%. It is clear that the lpsr phases of spots 1, 3, and 4 are different than those of spots 2 and 5 which their contents of $SiO_2$ and $Al_2O_3$ seem to be associated with the str and/or mol water characteristic for them.

The lpsr spots of these two last grains have contents of $TiO_2$, $Fe_2O_3$, Fe, and Ti/(Ti + Fe) ratios in the ranges of 73.63–83.14%, 18.73–11.62%, 0.97–0.5, and 0.75–0.86, respectively. The detected lpsr spots of the last three grains have the chemical formulas in the range of $Fe_{0.97-0.5}Ti_3O_{5.89-4.45}(OH)_{3.11-4.55}$ (Table 1).

The grain (Figure 2g) was originally a definite altered silicate mineral where most of the contents of $SiO_2$, $Al_2O_3$, $Fe_2O_3$, MgO, $Na_2O$, and $K_2O$ are leached out with the enrichment of the minor contained $TiO_2$. The detection of the various analyzed spots of the grain reflects that the decreasing content of these oxide contents almost equals the increasing content of $TiO_2$. Note that the chemical composition of spot 1 seems as if it was lpsr (Table 1, grain 2g).

Both the grains (Figure 2h,i) have a similar lpsr phase which was broken into oxy- and/or oxyhydroxide iron and titanium individual mineral phases. In the grain (Figure 2j), spots 1 and 2 are lpsr of a definite phase while spots 3 and 4 are lpsr of a different phase.

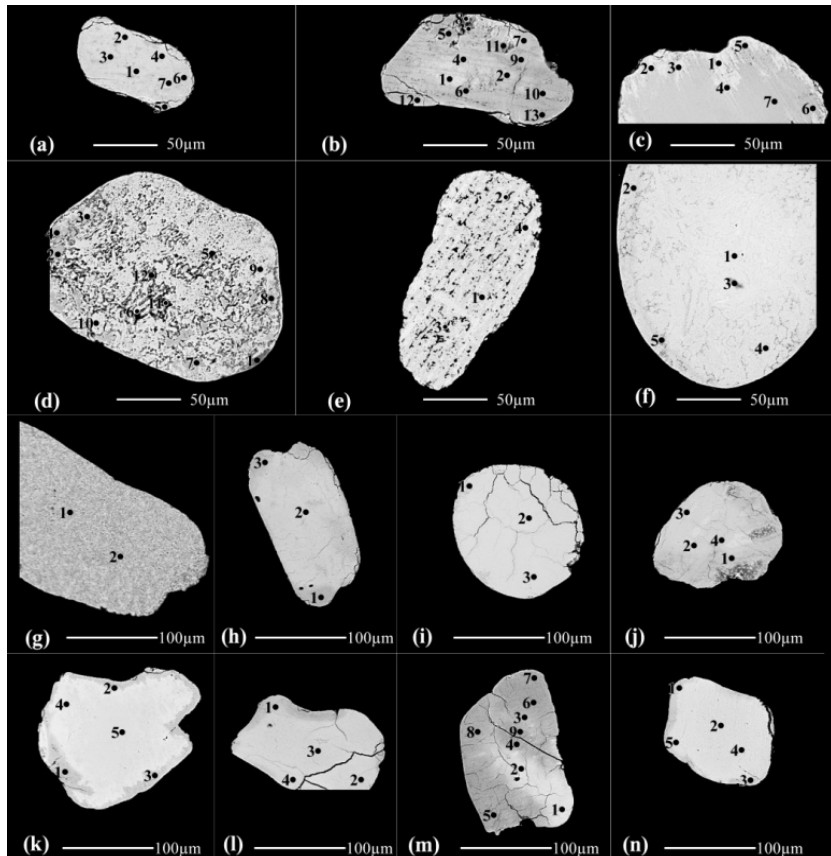

**Figure 2.** The backscattered electron (BSE) images for the altered brown, yellowish-brown, and brownish-yellow ilmenite grains (**a–n**), and the locations of their analyzed spots, separated as magnetic fraction at 0.5 ampere value.

Both the grains (Figure 2k,l) have collapsed lpsr into oxy- and/or oxyhydroxide iron and titanium individual phases. Comparing spots 1, 2, and 3 with spots 4 and 5 of grain 2k reflects the effect of the content of structural and/or molecular water on the darkness of the BSE image for the analyzed spots whereas their contents increase as the darkness degree increases.

In the grain (Figure 2m), analyzed spots 1,2, and 3 are a definite lpsr phase mixed with an individual Ti-bearing phase. Not all the analyzed $TiO_2$ content is included within the lpsr structure. Spots from 4 to 9 are composed of mixed mineral phases of individual iron and titanium which occur as oxy- and/or oxyhydroxides. The last conclusion is the result of comparing the BSE image for the analyzed spots of the grain (Figure 2m), and their corresponding OT% values (Table 1). The recorded lpsr spots of the two grains (Figure 2j,m) have contents of $TiO_2$, $Fe_2O_3$, Fe, and Ti/(Ti + Fe) ratiosin the ranges of 76.99–83.17%, 15.45–9.12%, 0.85–0.53, and 0.78–0.85, respectively. The detected lpsr spots of the grains have the chemical formulas in the range of $Fe_{0.85-0.53}Ti_3O_{5.54-4.58}(OH)_{3.46-4.42}$ (Figure 2g–m, Table 1).

In addition, the grain (Figure 2n, Table 1) is composed of mixed two individual phases of iron and titanium present most probably as oxy- and/or oxyhydroxides. Spot 1 contains more structural and/or molecular water although it has relatively lower $TiO_2$ content. All the detected lpsr spots of the grains (Figure 2a–n) have the chemical formulas in the range of $Fe_{1.49-0.50}Ti_3O_{7.4-4.45}(OH)_{1.6-3.46}$ (Table 1). The Ti/(Ti + Fe) ratio ranges between 0.67 and 0.86.

**Table 1.** The microprobe chemical analyses, the corresponding molecular formula and mineral phases of the analyzed spots of the altered brown ilmenite grains; (Figure 1a–g), the altered brown, yellowish brown&brownish yellow ilmenite grains; (Figure 2a–n), the altered pale brown, yellow, creamy and white coloured ilmenite grains; (Figure 3a–i), separated as magnetic at 0.5 ampere value. OT is the original analyzed total oxides sum while NT is the new one after applying the constructed Excel software and calculating the contained $H_2O$ wt%.

| Grains | Spots | Wt% | | | | | | | | | | | O Total | OH% | H₂O% | N Total | Molecular Formula | | | | Lost Fe | Ti/(Ti + Fe) | Mineral Phase |
|---|---|---|---|---|---|---|---|---|---|---|---|---|---|---|---|---|---|---|---|---|---|---|---|
| | | SiO₂ | MgO | MnO | CaO | ZnO | Fe₂O₃ | Al₂O₃ | Cr₂O₃ | Na₂O | K₂O | TiO₂ | | | | | Fe₂ | Ti₃ | Ox | OHy | | | |
| 1a | 1 | 0.62 | 0.26 | 0.57 | 0.46 | 0.00 | 20.70 | 0.63 | 0.11 | 0.02 | 0.01 | 68.81 | 92.19 | 13.94 | 7.38 | 99.58 | 1.07 | 3 | 6.16 | 2.84 | 0.93 | 0.74 | psr/lpsr |
| | 4 | 0.68 | 0.25 | 0.51 | 0.51 | 0.08 | 19.42 | 0.61 | 0.12 | 0.00 | 0.01 | 70.50 | 92.69 | 15.22 | 8.06 | 100.75 | 0.99 | 3 | 5.93 | 3.07 | 1.01 | 0.75 | psr/lpsr |
| | 7 | 0.75 | 0.26 | 0.59 | 0.39 | 0.16 | 18.60 | 0.67 | 0.11 | 0.00 | 0.01 | 71.42 | 92.98 | 15.87 | 8.40 | 101.38 | 0.95 | 3 | 5.82 | 3.18 | 1.05 | 0.76 | psr/lpsr |
| | 8 | 0.74 | 0.25 | 0.51 | 0.37 | 0.03 | 18.51 | 0.64 | 0.14 | 0.01 | 0.03 | 72.15 | 93.37 | 16.26 | 8.61 | 101.99 | 0.93 | 3 | 5.75 | 3.25 | 1.07 | 0.76 | psr/lpsr |
| 1b | 1 | 0.70 | 0.11 | 0.29 | 0.29 | 0.00 | 15.79 | 0.75 | 0.35 | 0.01 | 0.01 | 76.62 | 94.89 | 19.26 | 10.20 | 105.09 | 0.75 | 3 | 5.26 | 3.74 | 1.25 | 0.80 | psr/lpsr |
| | 2 | 0.74 | 0.12 | 0.20 | 0.32 | 0.00 | 14.22 | 1.01 | 0.30 | 0.00 | 0.01 | 76.88 | 93.80 | 20.17 | 10.68 | 104.49 | 0.70 | 3 | 5.11 | 3.89 | 1.30 | 0.81 | psr/lpsr |
| | 4 | 0.76 | 0.07 | 0.19 | 0.32 | 0.00 | 14.13 | 1.09 | 0.39 | 0.01 | 0.00 | 77.81 | 94.78 | 20.27 | 10.74 | 105.51 | 0.70 | 3 | 5.10 | 3.90 | 1.30 | 0.81 | psr/lpsr |
| | 8 | 0.90 | 0.11 | 0.11 | 0.32 | 0.06 | 12.72 | 1.13 | 0.45 | 0.04 | 0.04 | 79.01 | 94.89 | 21.20 | 11.23 | 106.11 | 0.65 | 3 | 4.96 | 4.04 | 1.35 | 0.82 | psr/lpsr |
| 1c | 1 | 48.27 | 0.70 | 0.01 | 0.20 | 0.04 | 3.49 | 36.23 | 0.03 | 0.04 | 0.25 | 1.82 | 91.07 | | | | | | | | | | Silicate mineral |
| | 2 | 0.92 | 0.17 | 0.09 | 0.39 | 0.03 | 7.78 | 1.62 | 0.63 | 0.05 | 0.03 | 82.15 | 93.83 | 24.53 | 12.99 | 106.82 | 0.49 | 3 | 4.46 | 4.54 | 1.51 | 0.86 | psr/lpsr |
| | 4 | 1.07 | 0.26 | 0.03 | 0.41 | 0.07 | 7.25 | 1.53 | 0.79 | 0.03 | 0.05 | 82.43 | 93.91 | 24.66 | 13.06 | 106.97 | 0.48 | 3 | 4.44 | 4.56 | 1.52 | 0.86 | psr/lpsr |
| | 8 | 0.87 | 0.24 | 0.06 | 0.37 | 0.14 | 6.68 | 1.40 | 0.70 | 0.03 | 0.04 | 84.11 | 94.65 | 25.67 | 13.60 | 108.24 | 0.43 | 3 | 4.29 | 4.71 | 1.57 | 0.87 | psr/lpsr |
| 1d | 1 | 5.85 | 0.93 | 0.15 | 0.50 | 0.02 | 16.57 | 2.68 | 0.46 | 0.05 | 0.29 | 67.89 | 95.38 | 7.46 | 3.95 | 99.34 | 1.43 | 3 | 7.46 | 1.54 | 0.57 | 0.68 | psr/lpsr |
| | 2 | 1.71 | 0.51 | 0.41 | 0.37 | 0.01 | 23.28 | 1.12 | 0.25 | 0.01 | 0.03 | 68.66 | 96.34 | 10.09 | 5.34 | 101.68 | 1.30 | 3 | 6.89 | 2.11 | 0.70 | 0.70 | psr/lpsr |
| | 3 | 0.83 | 0.40 | 0.20 | 0.36 | 0.00 | 22.93 | 0.82 | 0.32 | 0.02 | 0.05 | 69.50 | 95.43 | 12.05 | 6.38 | 101.81 | 1.18 | 3 | 6.51 | 2.49 | 0.82 | 0.72 | psr/lpsr |
| | 4 | 1.08 | 0.50 | 0.29 | 0.40 | 0.03 | 21.81 | 0.97 | 0.35 | 0.05 | 0.04 | 70.60 | 96.10 | 12.48 | 6.61 | 102.71 | 1.16 | 3 | 6.43 | 2.57 | 0.84 | 0.72 | psr/lpsr |
| | 7 | 1.49 | 0.21 | 0.01 | 0.54 | 0.02 | 9.70 | 1.22 | 0.60 | 0.00 | 0.06 | 81.86 | 95.69 | 22.79 | 12.07 | 107.76 | 0.57 | 3 | 4.73 | 4.27 | 1.43 | 0.84 | psr/lpsr |
| 1e | 1 | 46.02 | 6.23 | 0.09 | 0.62 | 0.10 | 16.12 | 10.21 | 0.03 | 0.10 | 3.82 | 7.53 | 90.85 | | | | | | | | | | Silicate mineral |
| | 5 | 35.55 | 6.48 | 0.17 | 0.44 | 0.01 | 18.82 | 8.20 | 0.03 | 0.06 | 2.18 | 22.43 | 94.38 | | | | | | | | | | Silicate mineral |
| | 6 | 1.60 | 0.87 | 0.43 | 0.14 | 0.07 | 27.95 | 0.55 | 0.10 | 0.03 | 0.11 | 64.32 | 96.15 | 6.04 | 3.20 | 99.35 | 1.58 | 3 | 7.69 | 1.31 | 0.42 | 0.66 | psr/lpsr |
| | 8 | 0.59 | 0.40 | 0.24 | 0.24 | 0.08 | 14.51 | 0.66 | 0.08 | 0.06 | 0.06 | 79.76 | 96.66 | 20.82 | 11.03 | 107.69 | 0.68 | 3 | 5.00 | 4.00 | 1.32 | 0.81 | psr/lpsr |
| | 9 | 0.16 | 0.13 | 0.10 | 0.23 | 0.03 | 3.18 | 0.93 | 0.04 | 0.06 | 0.01 | 91.11 | 95.97 | 30.76 | 16.29 | 112.26 | 0.19 | 3 | 3.54 | 5.46 | 1.81 | | TiO₂ and Fe₂O₃ |
| | 12 | 0.03 | 0.03 | 0.07 | 0.18 | 0.01 | 1.25 | 1.12 | 0.12 | 0.06 | 0.02 | 94.13 | 97.01 | 32.31 | 17.11 | 114.12 | 0.12 | 3 | 3.34 | 5.66 | 1.88 | | TiO₂ and Fe₂O₃ |
| 1f | 1 | 36.56 | 1.09 | 0.12 | 1.10 | 0.00 | 3.76 | 25.65 | 0.06 | 0.04 | 0.14 | 6.68 | 75.20 | | | | | | | | | | Silicate mineral |
| | 6 | 12.21 | 0.46 | 0.09 | 0.32 | 0.00 | 2.20 | 9.13 | 0.04 | 0.00 | 0.06 | 72.26 | 96.76 | | | | | | | | | | Silicate mineral |
| | 9 | 0.37 | 0.13 | 1.04 | 0.28 | 0.02 | 28.06 | 0.22 | 0.00 | 0.04 | 0.00 | 65.85 | 96.00 | 8.61 | 4.56 | 100.56 | 1.40 | 3 | 7.15 | 1.85 | 0.60 | 0.68 | psr/lpsr |
| | 13 | 0.42 | 0.08 | 0.94 | 0.32 | 0.08 | 26.55 | 0.16 | 0.05 | 0.02 | 0.00 | 67.53 | 96.14 | 10.20 | 5.40 | 101.54 | 1.30 | 3 | 6.84 | 2.16 | 0.70 | 0.70 | psr/lpsr |
| | 17 | 0.85 | 0.18 | 0.35 | 0.52 | 0.00 | 14.09 | 0.35 | 0.06 | 0.03 | 0.00 | 68.42 | 84.85 | 19.19 | 10.16 | 95.01 | 0.76 | 3 | 5.27 | 3.73 | 1.24 | 0.80 | psr/lpsr |
| | 21 | 0.01 | 0.01 | 0.25 | 0.04 | 0.01 | 13.60 | 0.00 | 0.03 | 0.00 | 0.01 | 86.56 | 100.49 | 24.26 | 12.85 | 113.34 | 0.49 | 3 | 4.44 | 4.56 | 1.51 | 0.86 | psr/lpsr |
| | 22 | 0.01 | 0.00 | 0.02 | 0.06 | 0.00 | 2.53 | 0.00 | 0.10 | 0.00 | 0.00 | 96.17 | 98.89 | 32.75 | 17.35 | 116.24 | 0.09 | 3 | 3.25 | 5.75 | 1.91 | | TiO₂ and Fe₂O₃ |
| | 24 | 0.28 | 0.00 | 0.01 | 0.04 | 0.01 | 0.44 | 0.17 | 0.10 | 0.02 | 0.01 | 98.97 | 100.04 | 33.83 | 17.92 | 117.96 | 0.04 | 3 | 3.13 | 5.87 | 1.96 | | Mainly TiO₂ |
| 1g | 1 | 0.20 | 0.09 | 0.67 | 0.13 | 0.00 | 28.69 | 0.22 | 0.02 | 0.01 | 0.01 | 65.75 | 95.78 | 8.69 | 4.60 | 100.38 | 1.39 | 3 | 7.13 | 1.87 | 0.61 | 0.68 | psr/lpsr |
| | 10 | 0.40 | 0.13 | 0.53 | 0.31 | 0.00 | 24.98 | 0.40 | 0.08 | 0.03 | 0.03 | 68.71 | 95.59 | 11.60 | 6.14 | 101.73 | 1.21 | 3 | 6.58 | 2.42 | 0.79 | 0.71 | psr/lpsr |
| | 11 | 0.41 | 0.10 | 0.53 | 0.32 | 0.03 | 23.78 | 0.38 | 0.05 | 0.02 | 0.01 | 69.93 | 95.56 | 12.81 | 6.79 | 102.34 | 1.13 | 3 | 6.35 | 2.65 | 0.87 | 0.73 | psr/lpsr |
| | 12 | 0.54 | 0.16 | 0.39 | 0.38 | 0.07 | 22.09 | 0.46 | 0.04 | 0.01 | 0.01 | 70.46 | 94.60 | 13.90 | 7.36 | 101.96 | 1.06 | 3 | 6.16 | 2.84 | 0.94 | 0.74 | psr/lpsr |
| | 13 | 0.69 | 0.16 | 0.32 | 0.36 | 0.00 | 21.75 | 0.71 | 0.13 | 0.03 | 0.02 | 70.65 | 94.82 | 13.76 | 7.29 | 102.11 | 1.07 | 3 | 6.19 | 2.81 | 0.93 | 0.74 | psr/lpsr |
| | 15 | 1.04 | 0.21 | 0.13 | 0.45 | 0.14 | 13.40 | 1.12 | 0.12 | 0.05 | 0.03 | 75.04 | 91.71 | 19.90 | 10.54 | 102.25 | 0.73 | 3 | 5.17 | 3.83 | 1.27 | 0.81 | psr/lpsr |
| | 17 | 1.08 | 0.22 | 0.15 | 0.52 | 0.07 | 11.13 | 1.09 | 0.04 | 0.10 | 0.02 | 78.27 | 92.68 | 22.11 | 11.71 | 104.38 | 0.61 | 3 | 4.82 | 4.18 | 1.39 | 0.83 | psr/lpsr |
| | 19 | 1.20 | 0.21 | 0.13 | 0.57 | 0.10 | 9.14 | 1.17 | 0.02 | 0.07 | 0.00 | 79.66 | 92.24 | 23.59 | 12.50 | 104.74 | 0.54 | 3 | 4.60 | 4.40 | 1.46 | 0.85 | psr/lpsr |
| | 20 | 0.08 | 0.08 | 0.01 | 0.12 | 0.09 | 0.79 | 1.07 | 0.03 | 0.08 | 0.04 | 93.80 | 96.19 | 32.66 | 17.30 | 113.49 | 0.11 | 3 | 3.29 | 5.71 | 1.89 | | Mainly TiO₂ |

**Table 1.** *Cont.*

| Grains | Spots | SiO$_2$ | MgO | MnO | CaO | ZnO | Fe$_2$O$_3$ | Al$_2$O$_3$ | Cr$_2$O$_3$ | Na$_2$O | K$_2$O | TiO$_2$ | O Total | OH% | H$_2$O% | N Total | Fe$_2$ | Ti$_3$ | Ox | OHy | Lost Fe | Ti/(Ti + Fe) | Mineral Phase |
|---|---|---|---|---|---|---|---|---|---|---|---|---|---|---|---|---|---|---|---|---|---|---|---|
| | | | | | | | | | Wt% | | | | | | | | | Molecular Formula | | | | | |
| 2a | 1 | 1.03 | 0.72 | 0.34 | 0.31 | 0.07 | 20.14 | 0.97 | 0.18 | 0.01 | 0.03 | 72.24 | 96.04 | 14.09 | 7.46 | 103.50 | 1.06 | 3 | 6.15 | 2.85 | 0.94 | 0.74 | psr/lpsr |
| | 4 | 1.02 | 0.75 | 0.40 | 0.34 | 0.03 | 19.66 | 1.01 | 0.27 | 0.05 | 0.04 | 73.30 | 96.86 | 14.51 | 7.68 | 104.54 | 1.04 | 3 | 6.07 | 2.93 | 0.96 | 0.74 | psr/lpsr |
| | 7 | 2.32 | 0.47 | 0.17 | 0.46 | 0.10 | 11.24 | 1.50 | 0.30 | 0.03 | 0.11 | 79.70 | 96.37 | 20.01 | 10.60 | 106.97 | 0.72 | 3 | 5.18 | 3.82 | 1.28 | 0.81 | psr/lpsr |
| 2b | 1 | 0.90 | 0.42 | 0.52 | 0.21 | 0.00 | 24.15 | 1.18 | 0.38 | 0.00 | 0.03 | 68.83 | 96.61 | 10.44 | 5.53 | 102.14 | 1.28 | 3 | 6.81 | 2.19 | 0.72 | 0.70 | psr/lpsr |
| | 2 | 0.73 | 0.25 | 0.20 | 0.27 | 0.02 | 17.42 | 1.76 | 0.62 | 0.00 | 0.02 | 72.10 | 93.38 | 15.78 | 8.36 | 101.74 | 0.95 | 3 | 5.86 | 3.14 | 1.05 | 0.76 | psr/lpsr |
| | 3 | 0.75 | 0.22 | 0.03 | 0.29 | 0.00 | 6.15 | 3.17 | 0.77 | 0.02 | 0.02 | 72.84 | 84.25 | 23.07 | 12.22 | 96.47 | 0.57 | 3 | 4.71 | 4.29 | 1.43 | | TiO$_2$ and Fe$_2$O$_3$ |
| | 7 | 0.65 | 0.20 | 0.17 | 0.23 | 0.00 | 13.49 | 1.51 | 0.60 | 0.01 | 0.01 | 75.40 | 92.28 | 19.81 | 10.49 | 102.77 | 0.73 | 3 | 5.18 | 3.82 | 1.27 | 0.80 | psr/lpsr |
| | 8 | 1.14 | 0.27 | 0.05 | 0.30 | 0.00 | 8.29 | 5.29 | 0.67 | 0.17 | 0.01 | 76.40 | 92.60 | 19.08 | 10.11 | 102.70 | 0.80 | 3 | 5.37 | 3.63 | 1.20 | | TiO$_2$ and Fe$_2$O$_3$ |
| | 9 | 0.81 | 0.23 | 0.08 | 0.30 | 0.00 | 10.92 | 1.78 | 0.80 | 0.00 | 0.01 | 78.87 | 93.79 | 21.66 | 11.47 | 105.26 | 0.63 | 3 | 4.90 | 4.10 | 1.37 | | TiO$_2$ and Fe$_2$O$_3$ |
| | 13 | 0.78 | 0.17 | 0.08 | 0.35 | 0.00 | 4.37 | 1.02 | 0.92 | 0.01 | 0.01 | 84.68 | 92.39 | 27.96 | 14.81 | 107.20 | 0.32 | 3 | 3.95 | 5.05 | 1.68 | | TiO$_2$ and Fe$_2$O$_3$ |
| 2c | 2 | 0.59 | 0.13 | 1.01 | 0.28 | 0.06 | 25.11 | 0.57 | 0.46 | 0.00 | 0.01 | 68.25 | 96.45 | 10.46 | 5.54 | 101.99 | 1.28 | 3 | 6.79 | 2.21 | 0.72 | 0.70 | psr/lpsr |
| | 4 | 0.39 | 0.17 | 1.13 | 0.22 | 0.08 | 25.00 | 0.38 | 0.32 | 0.00 | 0.01 | 69.34 | 97.04 | 11.29 | 5.98 | 103.02 | 1.23 | 3 | 6.63 | 2.37 | 0.77 | 0.71 | psr/lpsr |
| | 6 | 1.14 | 0.19 | 0.23 | 0.50 | 0.00 | 13.05 | 1.01 | 1.04 | 0.00 | 0.01 | 74.97 | 92.13 | 19.47 | 10.31 | 102.44 | 0.74 | 3 | 5.24 | 3.76 | 1.26 | 0.80 | psr/lpsr |
| | 8 | 1.15 | 0.16 | 0.20 | 0.51 | 0.05 | 13.89 | 0.96 | 0.93 | 0.05 | 0.02 | 77.16 | 95.08 | 19.34 | 10.24 | 105.32 | 0.75 | 3 | 5.26 | 3.74 | 1.25 | 0.80 | psr/lpsr |
| 2d | 1 | 29.91 | 13.50 | 0.84 | 0.35 | 0.00 | 25.04 | 17.44 | 0.00 | 0.10 | 0.51 | 0.97 | 88.65 | | | | | | | | | | Silicate mineral |
| | 3 | 22.18 | 0.02 | 0.03 | 20.69 | 0.00 | 0.49 | 0.49 | 0.01 | 0.00 | 0.01 | 56.16 | 100.08 | | | | | | | | | | Sphene |
| | 6 | 1.72 | 0.23 | 0.92 | 1.70 | 0.00 | 2.52 | 3.73 | 0.09 | 0.19 | 0.11 | 74.06 | 85.27 | 23.03 | 12.20 | 97.47 | 0.62 | 3 | 4.74 | 4.26 | 1.38 | | TiO$_2$ and Fe$_2$O$_3$ |
| | 7 | 6.94 | 0.10 | 0.16 | 6.76 | 0.00 | 1.43 | 1.10 | 0.04 | 0.08 | 0.04 | 82.06 | 98.69 | | | | | | | | | | Sphene and TiO$_2$ |
| | 8 | 1.28 | 0.32 | 0.67 | 1.26 | 0.00 | 3.40 | 3.16 | 0.09 | 0.18 | 0.04 | 85.32 | 95.70 | 25.34 | 13.42 | 109.13 | 0.49 | 3 | 4.37 | 4.63 | 1.51 | | TiO$_2$ and Fe$_2$O$_3$ |
| | 12 | 1.19 | 0.12 | 0.27 | 1.02 | 0.00 | 1.21 | 1.71 | 0.06 | 0.14 | 0.06 | 91.73 | 97.52 | 29.59 | 15.67 | 113.19 | 0.26 | 3 | 3.74 | 5.26 | 1.74 | | TiO$_2$ and Fe$_2$O$_3$ |
| 2e | 1 | 0.34 | 0.57 | 0.43 | 0.15 | 0.00 | 17.35 | 0.71 | 0.14 | 0.07 | 0.03 | 74.75 | 94.53 | 17.93 | 9.50 | 104.02 | 0.85 | 3 | 5.47 | 3.53 | 1.15 | 0.78 | psr/lpsr |
| | 2 | 0.20 | 0.41 | 0.12 | 0.08 | 0.00 | 11.62 | 0.33 | 0.10 | 0.05 | 0.02 | 83.14 | 96.07 | 24.35 | 12.90 | 108.96 | 0.50 | 3 | 4.45 | 4.55 | 1.50 | 0.86 | psr/lpsr |
| | 3 | 1.04 | 0.09 | 0.03 | 0.10 | 0.00 | 2.87 | 2.04 | 0.10 | 0.06 | 0.04 | 84.43 | 90.78 | 28.62 | 15.16 | 105.94 | 0.29 | 3 | 3.89 | 5.11 | 1.71 | | TiO$_2$ and Fe$_2$O$_3$ |
| | 4 | 0.04 | 0.22 | 0.04 | 0.01 | 0.00 | 7.77 | 0.11 | 0.05 | 0.01 | 0.00 | 89.72 | 97.96 | 28.46 | 15.07 | 113.03 | 0.29 | 3 | 3.84 | 5.16 | 1.71 | | TiO$_2$ and Fe$_2$O$_3$ |
| 2f | 1 | 0.72 | 0.32 | 0.21 | 0.24 | 0.00 | 18.73 | 1.43 | 0.43 | 0.08 | 0.07 | 73.63 | 95.85 | 15.56 | 8.24 | 104.10 | 0.97 | 3 | 5.89 | 3.11 | 1.03 | 0.75 | psr/lpsr |
| | 2 | 0.91 | 0.26 | 0.08 | 0.37 | 0.00 | 13.60 | 1.70 | 0.44 | 0.07 | 0.02 | 73.64 | 91.09 | 18.87 | 9.99 | 101.08 | 0.79 | 3 | 5.34 | 3.66 | 1.21 | 0.79 | psr/lpsr |
| | 3 | 0.72 | 0.31 | 0.16 | 0.23 | 0.00 | 17.99 | 1.30 | 0.41 | 0.02 | 0.03 | 74.07 | 95.25 | 16.43 | 8.70 | 103.95 | 0.92 | 3 | 5.74 | 3.26 | 1.08 | 0.77 | psr/lpsr |
| | 4 | 0.68 | 0.31 | 0.18 | 0.30 | 0.00 | 15.91 | 1.21 | 0.39 | 0.05 | 0.02 | 74.11 | 93.15 | 18.04 | 9.55 | 102.71 | 0.83 | 3 | 5.47 | 3.53 | 1.17 | 0.78 | psr/lpsr |
| | 5 | 1.01 | 0.25 | 0.08 | 0.44 | 0.00 | 12.30 | 1.73 | 0.52 | 0.08 | 0.03 | 74.47 | 90.89 | 19.76 | 10.46 | 101.35 | 0.74 | 3 | 5.20 | 3.80 | 1.26 | 0.80 | psr/lpsr |
| 2g | 1 | 13.38 | 1.22 | 0.03 | 0.43 | 0.00 | 8.71 | 4.02 | 0.10 | 0.14 | 0.72 | 68.87 | 97.62 | 2.91 | 1.54 | 99.16 | 1.64 | 3 | 8.41 | 0.59 | 0.36 | | TiO$_2$ + Silicate mineral |
| | 2 | 5.45 | 0.44 | 0.01 | 0.39 | 0.00 | 6.55 | 1.44 | 0.09 | 0.19 | 0.28 | 82.75 | 97.59 | 20.59 | 10.90 | 108.50 | 0.67 | 3 | 5.16 | 3.84 | 1.33 | | TiO$_2$ + Silicate mineral |
| 2h | 2 | 1.26 | 0.23 | 0.17 | 0.29 | 0.00 | 6.64 | 1.56 | 1.08 | 0.06 | 0.03 | 85.63 | 96.95 | 25.05 | 13.27 | 110.21 | 0.46 | 3 | 4.39 | 4.61 | 1.54 | | TiO$_2$ and Fe$_2$O$_3$ |
| 2i | 1 | 0.99 | 0.19 | 0.17 | 0.41 | 0.00 | 5.56 | 1.03 | 1.51 | 0.05 | 0.00 | 84.84 | 94.74 | 26.25 | 13.90 | 108.64 | 0.40 | 3 | 4.20 | 4.80 | 1.60 | | TiO$_2$ and Fe$_2$O$_3$ |
| | 3 | 1.01 | 0.16 | 0.08 | 0.37 | 0.00 | 5.55 | 1.04 | 1.62 | 0.10 | 0.05 | 87.32 | 97.28 | 26.42 | 13.99 | 111.27 | 0.39 | 3 | 4.17 | 4.83 | 1.61 | | TiO$_2$ and Fe$_2$O$_3$ |
| 2j | 2 | 0.94 | 0.54 | 0.09 | 0.46 | 0.00 | 13.33 | 0.74 | 0.91 | 0.02 | 0.05 | 79.56 | 96.64 | 20.42 | 10.82 | 107.45 | 0.70 | 3 | 5.08 | 3.92 | 1.30 | 0.81 | psr/lpsr |
| | 4 | 0.94 | 0.16 | 0.03 | 0.37 | 0.00 | 5.30 | 0.62 | 1.37 | 0.08 | 0.04 | 88.36 | 97.27 | 27.41 | 14.51 | 111.78 | 0.34 | 3 | 4.02 | 4.98 | 1.66 | | TiO$_2$ and Fe$_2$O$_3$ |
| 2k | 1 | 0.61 | 0.17 | 0.03 | 0.42 | 0.00 | 3.75 | 0.72 | 0.90 | 0.05 | 0.03 | 87.49 | 94.16 | 29.09 | 15.40 | 109.57 | 0.27 | 3 | 3.78 | 5.22 | 1.73 | | TiO$_2$ and Fe$_2$O$_3$ |
| | 2 | 0.62 | 0.15 | 0.03 | 0.32 | 0.00 | 4.12 | 0.67 | 0.73 | 0.05 | 0.05 | 87.88 | 94.63 | 29.07 | 15.40 | 110.02 | 0.27 | 3 | 3.78 | 5.22 | 1.73 | | TiO$_2$ and Fe$_2$O$_3$ |
| | 3 | 0.65 | 0.12 | 0.02 | 0.33 | 0.00 | 4.02 | 0.67 | 0.80 | 0.02 | 0.06 | 88.98 | 95.68 | 29.17 | 15.45 | 111.13 | 0.26 | 3 | 3.77 | 5.23 | 1.74 | | TiO$_2$ and Fe$_2$O$_3$ |
| | 4 | 0.86 | 0.15 | 0.00 | 0.42 | 0.00 | 2.98 | 0.73 | 1.38 | 0.07 | 0.05 | 91.48 | 98.12 | 29.25 | 15.49 | 113.61 | 0.26 | 3 | 3.77 | 5.23 | 1.74 | | TiO$_2$ and Fe$_2$O$_3$ |
| | 5 | 0.77 | 0.17 | 0.02 | 0.41 | 0.00 | 2.74 | 0.67 | 1.41 | 0.00 | 0.03 | 91.72 | 97.93 | 29.60 | 15.68 | 113.61 | 0.24 | 3 | 3.71 | 5.29 | 1.76 | | TiO$_2$ and Fe$_2$O$_3$ |

**Table 1.** *Cont.*

| Grains | Spots | SiO₂ | MgO | MnO | CaO | ZnO | Fe₂O₃ | Al₂O₃ | Cr₂O₃ | Na₂O | K₂O | TiO₂ | O Total | OH% | H₂O% | N Total | Fe₂ | Ti₃ | Ox | OHy | Lost Fe | Ti/(Ti + Fe) | Mineral Phase |
|---|---|---|---|---|---|---|---|---|---|---|---|---|---|---|---|---|---|---|---|---|---|---|---|
| 2l | 1 | 1.05 | 0.20 | 0.03 | 0.42 | 0.00 | 3.39 | 2.06 | 1.06 | 0.04 | 0.05 | 88.22 | 96.52 | 27.40 | 14.51 | 111.03 | 0.35 | 3 | 4.06 | 4.94 | 1.65 | | TiO₂ and Fe₂O₃ |
| | 4 | 1.03 | 0.21 | 0.08 | 0.37 | 0.00 | 2.79 | 1.82 | 1.10 | 0.07 | 0.05 | 89.26 | 96.77 | 28.13 | 14.90 | 111.67 | 0.32 | 3 | 3.95 | 5.05 | 1.68 | | TiO₂ and Fe₂O₃ |
| 2m | 2 | 0.78 | 0.19 | 0.09 | 0.20 | 0.00 | 11.07 | 2.64 | 0.18 | 0.08 | 0.03 | 81.63 | 96.89 | 21.62 | 11.45 | 108.33 | 0.64 | 3 | 4.92 | 4.08 | 1.36 | 0.82 | psr/lpsr |
| | 3 | 0.73 | 0.17 | 0.02 | 0.25 | 0.00 | 9.12 | 2.21 | 0.19 | 0.06 | 0.04 | 83.17 | 95.95 | 23.74 | 12.57 | 108.52 | 0.53 | 3 | 4.58 | 4.42 | 1.47 | 0.85 | psr/lpsr |
| | 5 | 0.75 | 0.15 | 0.07 | 0.27 | 0.00 | 5.23 | 1.61 | 0.24 | 0.06 | 0.01 | 85.39 | 93.77 | 27.37 | 14.50 | 108.27 | 0.35 | 3 | 4.04 | 4.96 | 1.65 | | TiO₂ and Fe₂O₃ |
| | 9 | 0.67 | 0.14 | 0.06 | 0.26 | 0.00 | 5.01 | 1.55 | 0.24 | 0.05 | 0.02 | 87.35 | 95.36 | 27.86 | 14.76 | 110.11 | 0.33 | 3 | 3.97 | 5.03 | 1.67 | | TiO₂ and Fe₂O₃ |
| 2n | 1 | 0.90 | 0.17 | 0.05 | 0.41 | 0.00 | 4.62 | 1.05 | 0.59 | 0.03 | 0.02 | 85.82 | 93.66 | 27.90 | 14.78 | 108.44 | 0.32 | 3 | 3.96 | 5.04 | 1.68 | | TiO₂ and Fe₂O₃ |
| | 5 | 0.88 | 0.15 | 0.03 | 0.29 | 0.00 | 3.83 | 0.99 | 0.69 | 0.08 | 0.04 | 90.05 | 97.02 | 28.87 | 15.29 | 112.31 | 0.28 | 3 | 3.82 | 5.18 | 1.72 | | TiO₂ and Fe₂O₃ |
| 3a | 2 | 1.03 | 0.24 | 0.08 | 0.53 | 0.00 | 9.89 | 3.14 | 0.60 | 0.09 | 0.02 | 78.37 | 93.98 | 20.59 | 10.90 | 104.89 | 0.70 | 3 | 5.09 | 3.91 | 1.30 | 0.81 | psr/lpsr |
| | 3 | 1.00 | 0.22 | 0.06 | 0.51 | 0.00 | 9.18 | 2.98 | 0.60 | 0.07 | 0.02 | 79.27 | 93.91 | 21.50 | 11.39 | 105.30 | 0.65 | 3 | 4.95 | 4.05 | 1.35 | 0.82 | psr/lpsr |
| | 5 | 0.98 | 0.24 | 0.04 | 0.40 | 0.00 | 10.39 | 2.90 | 0.81 | 0.12 | 0.05 | 80.73 | 96.66 | 20.83 | 11.03 | 107.69 | 0.69 | 3 | 5.05 | 3.95 | 1.31 | 0.81 | psr/lpsr |
| | 6 | 0.96 | 0.25 | 0.03 | 0.36 | 0.00 | 10.51 | 2.62 | 0.90 | 0.10 | 0.07 | 81.31 | 97.09 | 21.11 | 11.18 | 108.27 | 0.67 | 3 | 5.00 | 4.00 | 1.33 | 0.82 | psr/lpsr |
| 3b | 1 | 1.87 | 0.48 | 0.09 | 0.50 | 0.00 | 4.19 | 2.30 | 0.82 | 0.06 | 0.05 | 82.13 | 92.49 | 24.89 | 13.18 | 105.67 | 0.48 | 3 | 4.45 | 4.55 | 1.52 | | TiO₂ and Fe₂O₃ |
| | 3 | 1.23 | 0.31 | 0.11 | 0.41 | 0.00 | 4.36 | 2.11 | 0.65 | 0.12 | 0.09 | 84.15 | 93.54 | 26.18 | 13.87 | 107.40 | 0.42 | 3 | 4.24 | 4.76 | 1.58 | | TiO₂ and Fe₂O₃ |
| | 5 | 1.70 | 0.39 | 0.03 | 0.39 | 0.00 | 3.83 | 2.17 | 0.97 | 0.12 | 0.08 | 86.61 | 96.28 | 25.98 | 13.76 | 110.04 | 0.43 | 3 | 4.28 | 4.72 | 1.57 | | TiO₂ and Fe₂O₃ |
| 3c | 1 | 97.64 | 0.00 | 0.00 | 0.01 | 0.00 | 0.02 | 0.01 | 0.03 | 0.00 | 0.01 | 1.14 | 98.86 | | | | | | | | | | Silica |
| | 2 | 0.73 | 0.13 | 0.05 | 0.31 | 0.00 | 3.22 | 1.16 | 1.75 | 0.07 | 0.01 | 88.11 | 95.54 | 28.38 | 15.03 | 110.57 | 0.30 | 3 | 3.89 | 5.11 | 1.70 | | TiO₂ and Fe₂O₃ |
| | 6 | 0.79 | 0.14 | 0.05 | 0.30 | 0.00 | 3.35 | 1.23 | 1.64 | 0.08 | 0.02 | 89.49 | 97.08 | 28.33 | 15.00 | 112.08 | 0.30 | 3 | 3.90 | 5.10 | 1.70 | | TiO₂ and Fe₂O₃ |
| 3d | 1 | 10.89 | 4.09 | 0.13 | 0.17 | 0.00 | 8.40 | 4.86 | 0.10 | 0.03 | 0.82 | 67.82 | 97.32 | | | | | | | | | | Silicate mineral + TiO₂ |
| | 2 | 9.28 | 3.73 | 0.07 | 0.28 | 0.00 | 7.78 | 4.79 | 0.11 | 0.08 | 0.61 | 70.15 | 96.87 | | | | | | | | | | Silicate mineral + TiO₂ |
| | 4 | 7.54 | 3.32 | 0.07 | 0.14 | 0.00 | 6.33 | 3.81 | 0.11 | 0.04 | 0.19 | 75.80 | 97.35 | | | | | | | | | | Silicate mineral + TiO₂ |
| | 7 | 10.33 | 0.51 | 0.05 | 0.16 | 0.00 | 3.47 | 0.77 | 0.10 | 0.06 | 0.27 | 83.36 | 99.09 | | | | | | | | | | Silicate mineral + TiO₂ |
| 3e | 1 | 3.48 | 0.30 | 0.01 | 0.30 | 0.00 | 4.49 | 5.10 | 0.43 | 0.09 | 0.08 | 83.47 | 97.76 | 20.76 | 11.00 | 108.76 | 0.68 | 3 | 5.15 | 3.85 | 1.32 | | TiO₂ and Fe₂O₃ |
| | 4 | 2.77 | 0.26 | 0.02 | 0.28 | 0.00 | 4.00 | 4.44 | 0.44 | 0.06 | 0.05 | 85.30 | 97.62 | 22.89 | 12.12 | 109.74 | 0.57 | 3 | 4.80 | 4.20 | 1.43 | | TiO₂ and Fe₂O₃ |
| | 6 | 1.78 | 0.20 | 0.00 | 0.30 | 0.00 | 4.40 | 3.82 | 0.42 | 0.02 | 0.03 | 86.42 | 97.41 | 24.54 | 12.99 | 110.40 | 0.49 | 3 | 4.52 | 4.48 | 1.51 | | TiO₂ and Fe₂O₃ |
| 3f | 2 | 19.68 | 1.66 | 0.04 | 0.39 | 0.00 | 5.88 | 12.99 | 0.11 | 0.09 | 0.18 | 54.08 | 95.09 | | | | | | | | | | Silicate mineral |
| | 3 | 14.47 | 0.92 | 0.05 | 0.36 | 0.00 | 5.97 | 10.83 | 0.11 | 0.12 | 0.08 | 63.31 | 96.21 | | | | | | | | | | Silicate mineral |
| | 5 | 4.67 | 0.35 | 0.03 | 0.45 | 0.00 | 7.81 | 3.00 | 0.11 | 0.11 | 0.04 | 80.91 | 97.48 | 18.86 | 9.99 | 107.47 | 0.76 | 3 | 5.44 | 3.56 | 1.24 | | TiO₂ and Fe₂O₃ |
| | 7 | 1.49 | 0.04 | 0.03 | 0.36 | 0.00 | 7.45 | 0.74 | 0.08 | 0.11 | 0.02 | 87.33 | 97.65 | 26.11 | 13.83 | 111.48 | 0.40 | 3 | 4.22 | 4.78 | 1.60 | | TiO₂ and Fe₂O₃ |
| 3g | 1 | 46.65 | 1.49 | 0.03 | 0.44 | 0.00 | 5.27 | 29.01 | 0.04 | 1.29 | 2.73 | 3.11 | 90.06 | | | | | | | | | | Silicate mineral |
| | 3 | 30.48 | 0.00 | 0.00 | 28.84 | 0.00 | 0.14 | 0.56 | 0.01 | 0.02 | 0.00 | 41.06 | 101.11 | | | | | | | | | | Sphene |
| | 5 | 30.06 | 0.00 | 0.03 | 28.43 | 0.00 | 0.21 | 0.78 | 0.03 | 0.01 | 0.01 | 41.21 | 100.76 | | | | | | | | | | Sphene |
| | 6 | 5.10 | 0.00 | 0.00 | 0.46 | 0.00 | 0.18 | 0.09 | 0.06 | 0.00 | 0.03 | 84.87 | 90.80 | 27.97 | 14.81 | 105.61 | 0.28 | 3 | 4.05 | 4.95 | 1.72 | | TiO₂ and Fe₂O₃ |
| | 9 | 0.05 | 0.02 | 0.00 | 0.32 | 0.00 | 0.27 | 0.00 | 0.05 | 0.01 | 0.03 | 100.31 | 101.05 | 34.20 | 18.11 | 119.16 | 0.03 | 3 | 3.07 | 5.93 | 1.97 | | TiO₂ and Fe₂O₃ |

**Table 1.** *Cont.*

| Grains | Spots | SiO₂ | MgO | MnO | CaO | ZnO | Fe₂O₃ | Al₂O₃ | Cr₂O₃ | Na₂O | K₂O | TiO₂ | O Total | OH% | H₂O% | N Total | Fe₂ | Ti₃ | Ox | OHy | Lost Fe | Ti/(Ti + Fe) | Mineral Phase |
|---|---|---|---|---|---|---|---|---|---|---|---|---|---|---|---|---|---|---|---|---|---|---|---|
| | | | | | | | | | | | | | | | Wt% | | | | | | | | Molecular Formula |
| 3h | 3 | 28.34 | 0.01 | 0.00 | 26.51 | 0.00 | 1.11 | 2.44 | 0.01 | 0.00 | 0.01 | 41.92 | 100.33 | | | | | | | | | | Sphene |
| | 4 | 27.51 | 0.02 | 0.00 | 25.23 | 0.00 | 2.31 | 1.18 | 0.00 | 0.00 | 0.00 | 43.76 | 100.00 | | | | | | | | | | Sphene |
| | 5 | 2.04 | 0.38 | 0.11 | 0.99 | 0.00 | 3.33 | 8.81 | 0.02 | 0.18 | 0.06 | 72.12 | 88.04 | 16.90 | 8.95 | 96.99 | 0.94 | 3 | 5.81 | 3.19 | 1.06 | | TiO₂ and Fe₂O₃ |
| | 6 | 0.20 | 0.02 | 0.00 | 0.71 | 0.00 | 2.33 | 0.44 | 0.00 | 0.00 | 0.02 | 96.97 | 100.68 | 31.91 | 16.90 | 117.58 | 0.13 | 3 | 3.38 | 5.62 | 1.87 | | TiO₂ and Fe₂O₃ |
| | 10 | 0.01 | 0.01 | 0.00 | 0.56 | 0.00 | 0.35 | 0.00 | 0.01 | 0.02 | 0.01 | 100.07 | 101.02 | 34.07 | 18.04 | 119.06 | 0.04 | 3 | 3.08 | 5.92 | 1.96 | | TiO₂ and Fe₂O₃ |
| 3i | 1 | 0.51 | 0.10 | 0.03 | 0.24 | 0.00 | 5.97 | 0.87 | 0.78 | 0.05 | 0.00 | 84.72 | 93.27 | 27.49 | 14.56 | 107.83 | 0.34 | 3 | 4.01 | 4.99 | 1.66 | | TiO₂ and Fe₂O₃ |
| | 4 | 0.48 | 0.09 | 0.00 | 0.22 | 0.00 | 5.97 | 0.88 | 0.79 | 0.04 | 0.01 | 86.01 | 94.49 | 27.64 | 14.64 | 109.13 | 0.33 | 3 | 3.98 | 5.02 | 1.67 | | TiO₂ and Fe₂O₃ |
| | 8 | 0.51 | 0.08 | 0.00 | 0.21 | 0.00 | 5.59 | 0.81 | 0.79 | 0.11 | 0.03 | 89.22 | 97.36 | 28.16 | 14.91 | 112.27 | 0.31 | 3 | 3.91 | 5.09 | 1.69 | | TiO₂ and Fe₂O₃ |
| | 10 | 0.60 | 0.10 | 0.01 | 0.10 | 0.00 | 4.91 | 0.82 | 0.83 | 0.11 | 0.03 | 90.30 | 97.80 | 28.64 | 15.17 | 112.97 | 0.28 | 3 | 3.84 | 5.16 | 1.72 | | TiO₂ and Fe₂O₃ |

The investigation of the BSE images for many grains (Figure 2h–n) shows a surficial micro-structure similar to that which is called "shrinkage cracks", described by [25]. The same phenomenon can be noticed in the grain of Figure 1c. In fact, most of these grains are composed of individual phases of iron and titanium after the collapse of preexisting lpsr phases. This process is most probably associated with the losing of structural water and the formation of the more compact rutile structure and hence, the appearance of shrinkage cracks.

### 3.1.3. The Detected Pale Brown-, Yellow-, and Cream-Coloured Grains at 0.5 Ampere

Nine grains of the coloured grains were investigated (Figure 3, Table 1).The grain (Figure 3a) is composed of a definite lpsr phase with variable amounts of structural and/or molecular water. Spots 1, 2, and 3 contain relatively higher amounts of str and/or mol water than those of spots 4, 5, and 6. The detected lpsr spots have contents of $TiO_2$, $Fe_2O_3$, Fe, and Ti/(Ti + Fe) ratiosin the rangesof 77.38–81.31%, 10.89–9.18%, 0.72–0.67, and 0.81–0.82, respectively. The calculated chemical formulas areFe$_{0.72-0.67}$Ti$_3$O$_{5.14-4.95}$(OH)$_{3.86-4.05}$ (Table 1).

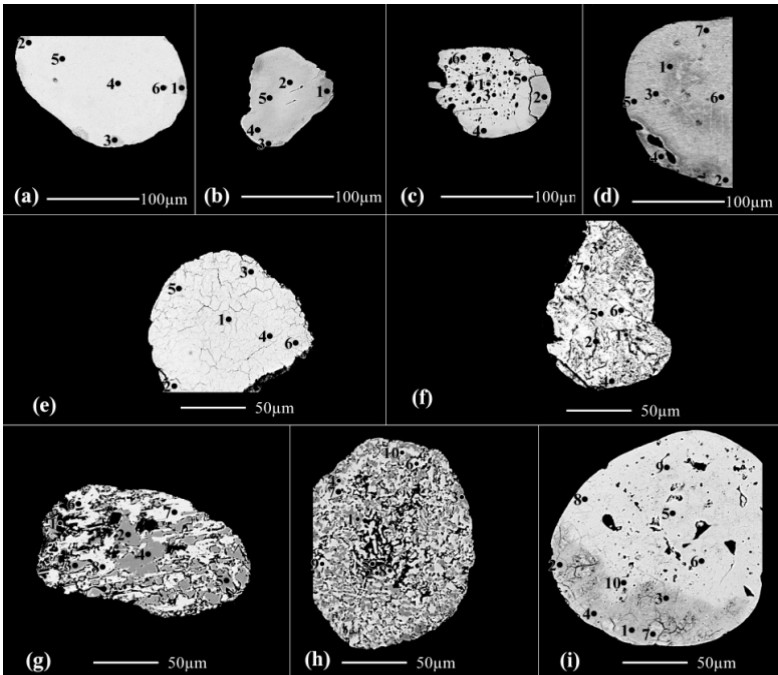

**Figure 3.** The backscattered electron (BSE) images for the altered pale brown-, yellow-, cream-, and white-coloured ilmenite grains (**a**–**i**), and the locations of their analyzed spots, separated as magnetic fraction at 0.5 ampere value.

Except for spot 1 of grain 3c, composed of silica, all three grains (Figure 3b,c,e) are individual phases of $TiO_2$ and $Fe_2O_3$ after the collapse of preexisting lpsr.

In the spots of the grain in Figure 3d, the contents of $Al_2O_3$ vary directly with both the contents of $Fe_2O_3$ and the contents of $SiO_2$, except spot 7. The values for OT of the various analyzed spots are more than 98%. It is clear that the contents of $SiO_2$, $Fe_2O_3$, and $Al_2O_3$ are related to an altered definite silicate mineral. Spot 7 may contain free silica in addition to the altered silicate mineral. As the essential contents of the silicate mineral are removed due to alteration, the contents of the $TiO_2$ are enriched (Table 1, grain 3d). In the grain in Figure 3f, spots 1, 2, and 3 are of a definite altered silicate mineral where most of $SiO_2$, $Al_2O_3$, and MgO contents are leached out while the $TiO_2$ contents are enriched. Spots 4, 5, 6, and 7 are mixed individual phases of Ti and Fe oxides.

Spot 1 of the grain (Figure 3g) is of a definite silicate mineral. Spots from 2 to 5 are sphene which seem to have a homogeneous chemical composition. Spots from 6 to 9 are for the individual $TiO_2$ phase, most probably rutile. The contents of $Al_2O_3$ in the last four

spots indicate that the $Al_2O_3$ are not compatible with the well-defined rutile structure; $TiO_2$ ranges between 84.87 and 100.31%.

Spots from 1 to 4 of the grain (Figure 3h) are sphene. Comparing the chemical composition of spots 3 and 4 with spot 5 may ensure that $SiO_2$ and CaO of sphene are leached by solutions containing both $Al_2O_3$ and $Na_2O$ in spot 5, leading to the enrichment of the immobile $TiO_2$. Spots from 6 to 10 are rutile after the leaching of sphene. These spots reflect the incompatibility of $SiO_2$ and $Al_2O_3$ in the structure of rutile (Table 1, grain 3h).

In the grain in Figure 3i, the spots of this grain are composed mainly of the individual $TiO_2$ phase and minor $Fe_2O_3$ phase due to the collapse of preexisting rutile. Additionally, note the shrinkage cracks of the two grains (Figure 3e,i).

### 3.1.4. The Detected Black Coloured Grains at 0.5 Ampere

Seven black grains were investigated (Figure 4, Table 2). The grain in Figure 4a is composed of Psr and lpsr phases. As the $TiO_2$ contents increase, the contents of structural and/or molecular water increase. In addition, according to the values for OT and NT of these spots, the calculated contents of structural water are not accepted. The presence of an individual $TiO_2$ phase associated with the lpsr phase is postulated (Table 2, grain 4a).

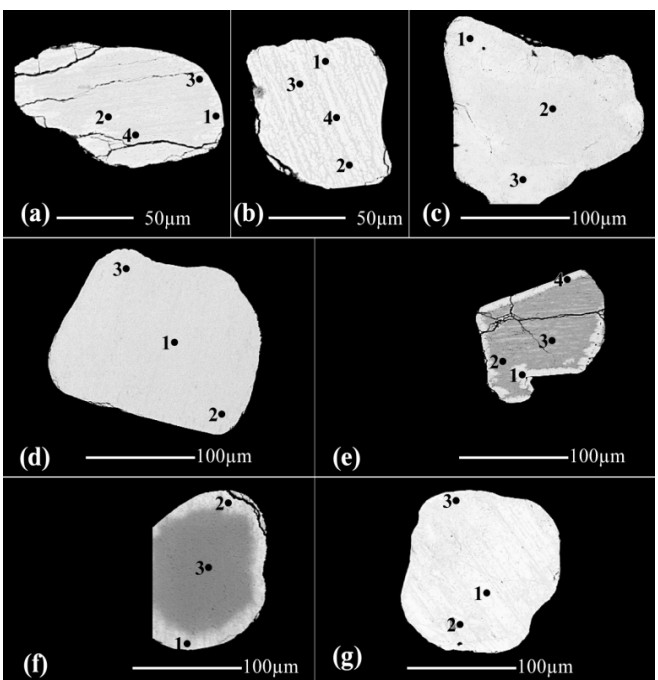

**Figure 4.** The backscattered electron (BSE) images for the alteredblack ilmenite grains; (**a**–**g**), and the locations of their analyzed spots, separated as magnetic fraction at 0.5 ampere value.

In the grain (Figure 4b), spot 1 is psr. Taking the calculated cationic iron of spot 1 into consideration (Table 2), it is clear that some of the analyzed iron is present in the divalent iron state $Fe^{++}$. The psr of spot 1 is different than those of spots 2 and 3 which in turn are different than that of spot 4.

The values for OT for the analyzed spots of the other five grains (Figure 4c–f) reflect that the grains are psr/lpsr of various chemical formulas. Some of their analyzed spots contain individual phases of $TiO_2$ in addition to the various lpsr phases. For example, spots 2, 3, and 4 of grain 4eare lpsr of the same phase. The calculated NT values ensure that the calculated structural water is incorrect where the mechanism of alteration at these $TiO_2$ contents is changed.

**Table 2.** The microprobe chemical analysis and the corresponding molecular formula of the analyzed spots of the altered black ilmenite grains; (Figure 4a–g), separated as magnetic fraction at 0.5 ampere value.

| Grains | Spots | Wt% | | | | | | | | | | | | | | | Molecular Formula | | | | Lost Fe | Ti/(Ti + Fe) | Mineral Phase |
| | | $SiO_2$ | MgO | MnO | CaO | ZnO | $Fe_2O_3$ | $Al_2O_3$ | $Cr_2O_3$ | $Na_2O$ | $K_2O$ | $TiO_2$ | O Total | OH% | $H_2O$% | N Total | $Fe_2$ | $Ti_3$ | Ox | OHy | | | |
|---|---|---|---|---|---|---|---|---|---|---|---|---|---|---|---|---|---|---|---|---|---|---|---|
| 4a | 1 | 0.44 | 0.18 | 0.93 | 0.20 | 0.00 | 29.35 | 0.26 | 0.19 | 0.04 | 0.03 | 64.15 | 95.76 | 6.97 | 3.69 | 99.45 | 1.51 | 3 | 7.48 | 1.52 | 0.49 | 0.66 | psr/lpsr |
| | 2 | 0.46 | 0.20 | 0.93 | 0.21 | 0.00 | 26.59 | 0.29 | 0.15 | 0.01 | 0.02 | 67.01 | 95.87 | 9.77 | 5.17 | 101.04 | 1.33 | 3 | 6.92 | 2.08 | 0.67 | 0.69 | psr/lpsr |
| | 3 | 0.58 | 0.22 | 0.80 | 0.24 | 0.00 | 24.25 | 0.35 | 0.23 | 0.09 | 0.04 | 68.44 | 95.23 | 11.54 | 6.11 | 101.35 | 1.22 | 3 | 6.59 | 2.41 | 0.78 | 0.71 | psr/lpsr |
| | 4 | 0.67 | 0.27 | 0.77 | 0.27 | 0.00 | 23.46 | 0.44 | 0.19 | 0.08 | 0.03 | 69.43 | 95.60 | 12.17 | 6.44 | 102.04 | 1.18 | 3 | 6.48 | 2.52 | 0.82 | 0.72 | psr/lpsr |
| 4b | 1 | 0.74 | 0.25 | 0.95 | 0.18 | 0.00 | 36.16 | 0.28 | 0.09 | 0.01 | 0.03 | 59.16 | 97.83 | 0.13 | 0.07 | 97.90 | 2.01 | 3 | 8.97 | 0.03 | -0.01 | 0.60 | psr/lpsr |
| | 2 | 0.27 | 0.14 | 0.86 | 0.14 | 0.00 | 32.45 | 0.18 | 0.10 | 0.00 | 0.00 | 63.08 | 97.22 | 4.96 | 2.63 | 99.85 | 1.65 | 3 | 7.89 | 1.11 | 0.35 | 0.65 | psr/lpsr |
| | 3 | 0.27 | 0.08 | 0.71 | 0.16 | 0.00 | 30.98 | 0.16 | 0.08 | 0.01 | 0.02 | 63.76 | 96.22 | 6.34 | 3.36 | 99.58 | 1.55 | 3 | 7.60 | 1.40 | 0.45 | 0.66 | psr/lpsr |
| | 4 | 0.61 | 0.19 | 0.44 | 0.25 | 0.00 | 25.14 | 0.36 | 0.14 | 0.04 | 0.03 | 68.88 | 96.07 | 11.32 | 6.00 | 102.06 | 1.22 | 3 | 6.63 | 2.37 | 0.78 | 0.71 | psr/lpsr |
| 4c | 1 | 0.12 | 0.07 | 1.50 | 0.08 | 0.00 | 34.40 | 0.11 | 0.02 | 0.03 | 0.00 | 62.68 | 99.01 | 3.57 | 1.89 | 100.90 | 1.76 | 3 | 8.19 | 0.81 | 0.24 | 0.63 | psr/lpsr |
| | 2 | 0.12 | 0.12 | 1.48 | 0.08 | 0.00 | 33.86 | 0.11 | 0.02 | 0.01 | 0.02 | 63.10 | 98.92 | 4.06 | 2.15 | 101.07 | 1.73 | 3 | 8.08 | 0.92 | 0.27 | 0.63 | psr/lpsr |
| | 3 | 0.17 | 0.11 | 1.62 | 0.11 | 0.00 | 33.97 | 0.12 | 0.03 | 0.02 | 0.01 | 63.68 | 99.83 | 4.05 | 2.15 | 101.97 | 1.73 | 3 | 8.09 | 0.91 | 0.27 | 0.63 | psr/lpsr |
| 4d | 1 | 0.20 | 0.30 | 0.93 | 0.07 | 0.00 | 31.97 | 0.15 | 0.05 | 0.00 | 0.01 | 63.75 | 97.43 | 5.54 | 2.93 | 100.36 | 1.61 | 3 | 7.77 | 1.23 | 0.39 | 0.65 | psr/lpsr |
| | 2 | 0.28 | 0.39 | 1.51 | 0.10 | 0.00 | 30.44 | 0.15 | 0.04 | 0.02 | 0.02 | 64.02 | 96.96 | 6.17 | 3.27 | 100.23 | 1.58 | 3 | 7.64 | 1.36 | 0.42 | 0.65 | psr/lpsr |
| | 3 | 0.35 | 0.32 | 0.43 | 0.12 | 0.00 | 29.03 | 0.20 | 0.05 | 0.08 | 0.01 | 66.45 | 97.07 | 8.37 | 4.43 | 101.50 | 1.42 | 3 | 7.20 | 1.80 | 0.58 | 0.68 | psr/lpsr |
| 4e | 1 | 0.20 | 0.21 | 1.40 | 0.12 | 0.00 | 31.33 | 0.14 | 0.04 | 0.06 | 0.02 | 65.10 | 98.63 | 6.22 | 3.29 | 101.92 | 1.58 | 3 | 7.63 | 1.37 | 0.42 | 0.66 | psr/lpsr |
| | 2 | 0.57 | 0.32 | 0.99 | 0.31 | 0.00 | 20.69 | 0.41 | 0.15 | 0.05 | 0.02 | 71.12 | 94.64 | 14.56 | 7.71 | 102.35 | 1.04 | 3 | 6.04 | 2.96 | 0.96 | 0.74 | psr/lpsr |
| | 3 | 0.61 | 0.29 | 1.10 | 0.28 | 0.00 | 20.31 | 0.47 | 0.16 | 0.08 | 0.03 | 71.48 | 94.81 | 14.76 | 7.82 | 102.63 | 1.03 | 3 | 6.01 | 2.99 | 0.97 | 0.74 | psr/lpsr |
| | 4 | 0.51 | 0.26 | 1.21 | 0.29 | 0.00 | 20.74 | 0.36 | 0.14 | 0.07 | 0.03 | 72.51 | 96.12 | 14.91 | 7.90 | 104.02 | 1.02 | 3 | 5.97 | 3.03 | 0.98 | 0.75 | psr/lpsr |
| 4f | 1 | 0.34 | 0.29 | 0.32 | 0.13 | 0.00 | 24.03 | 1.32 | 0.31 | 0.04 | 0.02 | 71.62 | 98.41 | 12.10 | 6.41 | 104.82 | 1.18 | 3 | 6.50 | 2.50 | 0.82 | 0.72 | psr/lpsr |
| | 2 | 0.47 | 0.25 | 0.21 | 0.16 | 0.00 | 21.62 | 1.46 | 0.45 | 0.04 | 0.01 | 74.70 | 99.38 | 14.20 | 7.52 | 106.89 | 1.05 | 3 | 6.12 | 2.88 | 0.95 | 0.74 | psr/lpsr |
| | 3 | 0.97 | 0.16 | 0.03 | 0.35 | 0.00 | 7.45 | 1.72 | 0.95 | 0.08 | 0.06 | 85.65 | 97.42 | 24.80 | 13.14 | 110.55 | 0.48 | 3 | 4.42 | 4.58 | 1.52 | 0.86 | psr/lpsr |
| 4g | 1 | 0.42 | 0.25 | 0.42 | 0.14 | 0.00 | 31.32 | 0.25 | 0.16 | 0.04 | 0.01 | 64.26 | 97.25 | 5.98 | 3.17 | 100.42 | 1.57 | 3 | 7.68 | 1.32 | 0.43 | 0.66 | psr/lpsr |
| | 2 | 0.65 | 0.23 | 0.25 | 0.22 | 0.00 | 27.76 | 0.42 | 0.27 | 0.05 | 0.03 | 67.82 | 97.69 | 9.08 | 4.81 | 102.50 | 1.36 | 3 | 7.06 | 1.94 | 0.64 | 0.69 | psr/lpsr |
| | 3 | 0.66 | 0.16 | 0.34 | 0.27 | 0.00 | 25.37 | 0.50 | 0.29 | 0.05 | 0.05 | 69.50 | 97.18 | 11.07 | 5.86 | 103.05 | 1.24 | 3 | 6.68 | 2.32 | 0.76 | 0.71 | psr/lpsr |

The detected psr/lpsr spots of the grains of (Figure 4) have contents of $TiO_2$, $Fe_2O_3$, Fe, and Ti/(Ti + Fe) ratios in the ranges of 59.16–74.7%, 36.16–20.31%, 2.01–0.48, and 0.6–0.75, respectively. The psr/lpsr spots of the grains have the chemical formulas in the range of $Fe_{2.01-0.48}Ti_3O_{8.97-4.42}(OH)_{0.03-4.48}$ (Table 2).

According to [36], within the pseudorutile composition range (60–71% $TiO_2$), the water content increased from ~2 to 4.5 wt.% with decreasing iron oxide content. The intermediate alteration phases comprised mixtures of $TiO_2$ with iron hydroxides.

In fact, within the given composition range of [36], the maximum limit of water content for the produced lpsr phases may be much more than 4.5 wt%.Even the lower limit value may be relatively lower than ~2%.

### 3.2. The Separated Magnetic Fraction at 1 Ampere

The magnetic fraction at one ampere is composed of light brown-, brownish yellow-, and cream-coloured grains in addition to minor amounts of the black grains. Most of the grains are spherical and sub-rounded to well rounded. The relatively coarser grains have highly pitted surfaces while the relatively finer ones have smooth surfaces.

Some grains of the magnetic fraction at 1 A are separated as light fraction of Clerici's solution (sp. Gr. = 4 g/cm$^3$). The grains have several coloured tints of pale brown, yellow, and cream with highly pitted surfaces. They contain a considerable number of the forementioned stained and coated grains.

Eight grains were investigated in the fraction (Figure 5, Table 3). Both of the two grains (Figure 5a,b) are black, the five grains (Figure 5c–g) have black cores and their surfaces are highly stained or partially coated with brownish, reddish, or yellowish white materials.The grain (Figure 5h), seems like locked grain between submetallic opaque mineral and altered light-coloured material.

The investigation of the analyzed spots of grain 5a (Table 3) shows that as the $TiO_2$ contents increase, the contents of str and/or mol water increase. In addition, some of the individual $TiO_2$ phase, most probably rutile, may be present with lpsr. The lpsr phase of spots 1 and 2 is different than that of spots 3 and 4.

The grain (Figure 5b) is composed of lpsr except for spot 1 which is leached ilmenite containing 38.6% $Fe_2O_3$, 0.51% FeO, and a sum total oxides of 99.89%. The calculated chemical formula is $Fe^{2+}_{0.03}Fe^{3+}_{1.94}$, other cations are $_{0.06}Ti_3O_9$ with a lost cationic iron amount of 0.97.

In the brownish black grain (Figure 5c), spot 1 is an inclusion of titanomagnetite. Spots from 2 to 5 are sphene. Spots from 8 to 12 are ferriferous rutile where the contents of $Fe_2O_3$ decrease as the contents of $TiO_2$ increase and the OT values range between 100.3 and 100.8%. According to the BSE image of spot 6, it most probably a mixture of hydrated Ti and Fe oxide phases. On the other hand, spot 7 is ferriferous rutile where most of its iron content is leached due to its location inside a void..

The spots of grain (Figure 5d) contain an enriched $TiO_2$ phase, most probably rutile, after the alteration of a definite silicate mineral composed mainly of $SiO_2$, $Al_2O_3$, $Fe_2O_3$, CaO, MgO, and $K_2O$ with minor $TiO_2$. Spots from 3 to 7 are a highly enriched $TiO_2$ phase after the leaching for most of the other associated oxides. It was observed that some spots of altered silicate minerals may be falsely considered as psr and lpsr according only to their chemical compositions. Hence, both of the back scattered electron images (BSE) of the analyzed spots and the chemical composition analyses are at least required for the correct interpretation for such investigated spots. In addition, it is obvious that the dependence on the powdered X-ray diffraction analysis (XRD) may not be enough to provide a correct decision during the investigation of some analyzed grains related to psr and lpsr alteration phases. The single crystal XRD may be a more efficient technique in such situations.

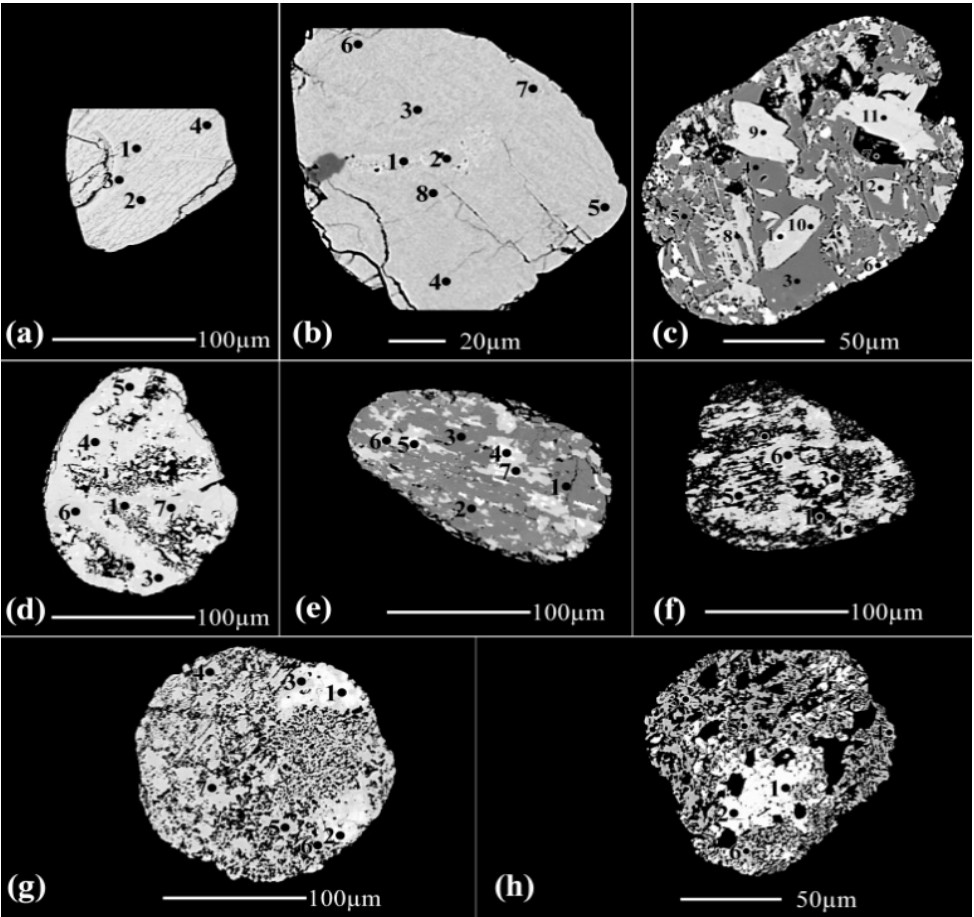

**Figure 5.** The backscattered electron (BSE) images for the altered black- and brownish black-coloured ilmenite grains (**a**–**h**), and the locations of their analyzed spots, separated as magnetic fraction at 1 ampere value.

In the grain in Figure 5e, spots 1, 2, and 3 are sphene, both of spots 4 and 5 are leached ilmenite where the calculated contents of FeO, $Fe_2O_3$, and lost iron are 8.21%, 19.99%; 29.59%, 21.04%; and 0.77, 0.55%, respectively. Both spots 6 and 7 are individual $TiO_2$ phases. Thus, it is clear that the preexisting ilmenite is partially substituted by sphene while the remaining is part altered to leached ilmenite.

Spot 1 of the grain in Figure 5f is an altered silicate mineral. Spot 2 is a mixture of silica and $TiO_2$. Spot 3 is ilmenite and rutile where most of the iron content is ferrous iron (Table 3); the other three spots are rutile.

Spots 1 and 2 of the grain in Figure 5g are psr; spot 3 is broken lpsr to rutile and hematite. Spots from 4 to 7 are rutile after the leaching of most of the associated $Fe_2O_3$ contents causing many of the scattered voids inside the grain.

Spots 1 and 2 of the grain in Figure 5h are leached ilmenite where the calculated contents of FeO, $Fe_2O_3$, and lost iron are 28.8%, 9.49%; 12.05%, 29.63%; and 0.33, 0.76, respectively. Spots from 3 to 6 are $TiO_2$ after leaching of the associated $Fe_2O_3$; both $TiO_2$ and $Fe_2O_3$ are obtained from the collapse of a preexisting lpsr phase.

The detected psr/lpsr spots of the grains in Figure 5a,b,g have contents of $TiO_2$, $Fe_2O_3$, Fe, and Ti/(Ti + Fe) ratios in the ranges of 60.75–79.91%, 37.3–11.47%, 1.86–0.62, and 0.62–0.83, respectively. The detected lpsr spots of all the grains of Figure 5 have the chemical formulas in the range of $Fe_{1.86-0.62}Ti_3O_{8.53-4.83}(OH)_{0.47-4.17}$ (Table 3).

**Table 3.** The microprobe chemical analyses and the corresponding molecular formula of each analyzed spot of the altered black and brownish black-opaque ilmenite grains; (Figure 5a–h) separated at 1 ampere value.

| Grains | Spots | SiO₂ | MgO | MnO | CaO | ZnO | Fe₂O₃ | Al₂O₃ | Cr₂O₃ | Na₂O | K₂O | TiO₂ | O Total | OH% | H₂O% | N Total | Fe₂ | Ti₃ | Ox | OHy | Losed Fe | Ti/(Ti + Fe) | Mineral Phase |
|---|---|---|---|---|---|---|---|---|---|---|---|---|---|---|---|---|---|---|---|---|---|---|---|
| 5a | 1 | 0.40 | 0.17 | 0.30 | 0.18 | 0.00 | 27.57 | 0.65 | 0.25 | 0.04 | 0.01 | 69.35 | 98.92 | 9.79 | 5.19 | 104.11 | 1.32 | 3 | 6.92 | 2.08 | 0.68 | 0.69 | psr/lpsr |
| | 3 | 0.71 | 0.25 | 0.21 | 0.42 | 0.00 | 13.63 | 1.14 | 0.47 | 0.05 | 0.05 | 78.89 | 95.81 | 20.48 | 10.84 | 106.65 | 0.70 | 3 | 5.07 | 3.93 | 1.30 | 0.81 | psr/lpsr |
| 5b | 1 | 0.13 | 0.06 | 0.47 | 0.05 | 0.00 | 39.18 | 0.06 | 0.03 | 0.07 | 0.02 | 59.87 | 99.96 | | | | | | | | | | Leached ilmenite |
| | 2 | 0.06 | 0.05 | 0.44 | 0.08 | 0.00 | 37.30 | 0.01 | 0.01 | 0.03 | 0.02 | 61.74 | 99.74 | 2.03 | 1.08 | 100.81 | 1.86 | 3 | 8.53 | 0.47 | 0.14 | 0.62 | psr/lpsr |
| | 8 | 0.23 | 0.06 | 0.60 | 0.11 | 0.00 | 29.71 | 0.24 | 0.07 | 0.03 | 0.03 | 66.31 | 97.39 | 8.13 | 4.30 | 101.69 | 1.43 | 3 | 7.24 | 1.76 | 0.57 | 0.68 | psr/lpsr |
| 5c | 1 | 0.00 | 0.00 | 0.00 | 0.22 | 0.00 | 98.84 | 0.00 | 0.04 | 0.01 | 0.00 | 4.34 | 103.45 | | | | | | | | | | Titanomagnetite |
| | 2 | 30.38 | 0.00 | 0.00 | 28.61 | 0.00 | 3.99 | 0.83 | 0.02 | 0.03 | 0.02 | 36.11 | 99.99 | | | | | | | | | | Sphene |
| | 5 | 28.26 | 0.02 | 0.16 | 26.60 | 0.00 | 1.62 | 0.31 | 0.03 | 0.02 | 0.04 | 43.84 | 100.90 | | | | | | | | | | Sphene |
| | 6 | 0.48 | 0.15 | 0.68 | 0.87 | 0.00 | 23.18 | 0.33 | 0.04 | 0.05 | 0.04 | 70.46 | 96.29 | 12.84 | 6.80 | 103.09 | 1.14 | 3 | 6.35 | 2.65 | 0.86 | | TiO₂ and Fe₂O₃ |
| | 7 | 1.91 | 0.28 | 0.10 | 1.53 | 0.00 | 4.04 | 1.28 | 0.11 | 0.25 | 0.21 | 81.81 | 91.51 | 25.77 | 13.65 | 105.15 | 0.46 | 3 | 4.29 | 4.71 | 1.54 | | TiO₂ and Fe₂O₃ |
| | 8 | 0.04 | 0.00 | 0.54 | 0.72 | 0.00 | 7.19 | 0.00 | 0.04 | 0.00 | 0.00 | 92.28 | 100.81 | | | | | | | | | | Ferriferous rutile |
| | 10 | 0.02 | 0.03 | 0.00 | 0.69 | 0.00 | 3.03 | 0.00 | 0.09 | 0.00 | 0.01 | 96.39 | 100.26 | | | | | | | | | | Ferriferous rutile |
| | 12 | 0.14 | 0.00 | 0.01 | 0.81 | 0.00 | 0.92 | 0.00 | 0.06 | 0.05 | 0.02 | 98.45 | 100.47 | | | | | | | | | | Ferriferous rutile |
| 5d | 3 | 0.03 | 0.03 | 0.00 | 0.01 | 0.00 | 1.35 | 0.00 | 0.02 | 0.01 | 0.02 | 98.09 | 99.55 | 33.68 | 17.84 | 117.38 | 0.05 | 3 | 3.14 | 5.86 | 1.95 | | TiO₂ and Fe₂O₃ |
| | 5 | 0.00 | 0.00 | 0.03 | 0.04 | 0.00 | 0.68 | 0.00 | 0.00 | 0.02 | 0.01 | 98.44 | 99.22 | 34.18 | 18.10 | 117.32 | 0.03 | 3 | 3.07 | 5.93 | 1.97 | | TiO₂ and Fe₂O₃ |
| | 7 | 0.11 | 0.00 | 0.00 | 0.05 | 0.00 | 0.40 | 0.00 | 0.00 | 0.04 | 0.02 | 99.68 | 100.30 | 34.26 | 18.14 | 118.44 | 0.02 | 3 | 3.06 | 5.94 | 1.98 | | TiO₂ and Fe₂O₃ |
| 5e | 1 | 30.09 | 0.00 | 0.02 | 28.55 | 0.00 | 0.88 | 1.18 | 0.07 | 0.03 | 0.03 | 38.02 | 98.86 | | | | | | | | | | Sphene |
| | 2 | 29.85 | 0.01 | 0.00 | 28.05 | 0.00 | 0.23 | 0.49 | 0.09 | 0.02 | 0.02 | 39.24 | 97.99 | | | | | | | | | | Sphene |
| | 3 | 30.06 | 0.00 | 0.00 | 28.13 | 0.00 | 0.32 | 0.28 | 0.08 | 0.04 | 0.03 | 40.95 | 99.89 | | | | | | | | | | Sphene |
| | 4 | 0.41 | 0.20 | 1.07 | 0.53 | 0.00 | 38.71 | 0.05 | 0.13 | 0.01 | 0.05 | 57.37 | 98.54 | | | | | | | | | | Leached ilmenite |
| | 5 | 0.00 | 0.16 | 2.26 | 0.42 | 0.00 | 43.24 | 0.00 | 0.12 | 0.01 | 0.01 | 57.48 | 103.71 | | | | | | | | | | Leached ilmenite |
| | 6 | 0.02 | 0.00 | 0.02 | 0.90 | 0.00 | 0.07 | 0.00 | 0.47 | 0.01 | 0.00 | 98.15 | 99.63 | 33.67 | 17.83 | 117.46 | 0.06 | 3 | 3.14 | 5.86 | 1.94 | | TiO₂ and Fe₂O₃ |
| | 7 | 0.00 | 0.00 | 0.00 | 0.62 | 0.00 | 0.18 | 0.00 | 0.44 | 0.00 | 0.00 | 98.36 | 99.61 | 33.84 | 17.92 | 117.53 | 0.05 | 3 | 3.11 | 5.89 | 1.95 | | TiO₂ and Fe₂O₃ |
| 5f | 1 | 43.69 | 4.87 | 0.05 | 0.42 | 0.00 | 15.28 | 11.64 | 0.05 | 0.15 | 3.26 | 9.04 | 88.45 | | | | | | | | | | Silicate mineral |
| | 2 | 78.87 | 0.95 | 0.00 | 0.16 | 0.00 | 2.19 | 2.07 | 0.03 | 0.10 | 0.59 | 10.83 | 95.79 | | | | | | | | | | Silica + TiO₂ |
| | 3 | 0.07 | 0.03 | 4.03 | 0.05 | 0.00 | 39.32 | 0.00 | 0.00 | 0.02 | 0.02 | 62.79 | 106.33 | | | | | | | | | | Ilmenite + TiO₂ |
| 5g | 1 | 0.95 | 0.37 | 0.72 | 0.33 | 0.00 | 31.19 | 0.82 | 0.30 | 0.10 | 0.02 | 60.75 | 95.54 | 3.01 | 1.60 | 97.14 | 1.80 | 3 | 8.32 | 0.68 | 0.20 | 0.63 | psr/lpsr |
| | 3 | 0.45 | 0.15 | 0.00 | 0.20 | 0.00 | 7.54 | 0.29 | 0.27 | 0.02 | 0.04 | 88.13 | 97.08 | 27.64 | 14.64 | 111.72 | 0.33 | 3 | 3.97 | 5.03 | 1.67 | | TiO₂ and Fe₂O₃ |
| | 4 | 0.40 | 0.05 | 0.00 | 0.07 | 0.00 | 1.64 | 0.13 | 0.04 | 0.00 | 0.01 | 97.41 | 99.76 | 32.88 | 17.41 | 117.17 | 0.08 | 3 | 3.25 | 5.75 | 1.92 | | TiO₂ and Fe₂O₃ |
| | 7 | 0.00 | 0.01 | 0.00 | 0.03 | 0.00 | 0.50 | 0.00 | 0.04 | 0.02 | 0.02 | 99.64 | 100.27 | 34.29 | 18.16 | 118.43 | 0.02 | 3 | 3.06 | 5.94 | 1.98 | | TiO₂ and Fe₂O₃ |
| 5h | 3 | 1.58 | 0.26 | 0.00 | 0.20 | 0.00 | 1.47 | 0.82 | 0.00 | 0.05 | 0.05 | 95.32 | 99.76 | 30.74 | 16.28 | 116.04 | 0.19 | 3 | 3.58 | 5.42 | 1.81 | | TiO₂ and Fe₂O₃ |
| | 6 | 0.52 | 0.01 | 0.02 | 0.13 | 0.00 | 0.85 | 0.40 | 0.02 | 0.01 | 0.01 | 97.26 | 99.24 | 33.04 | 17.50 | 116.73 | 0.08 | 3 | 3.24 | 5.76 | 1.92 | | TiO₂ and Fe₂O₃ |

### 3.3. The XRD Results

A small representative sample was obtained from the magnetic fraction at 0.50 ampere. A total of 200 grains were picked from each of the two detected coloured varieties, the various brown-coloured grainsand the black grains. The grains of each variety were split into two equal samples; each of 100 grains. One sample was subjected to the XRD and the other was roasted at 1100 °C for one hour and then treated using the XRD instrument.

The purpose of roasting the altered grains at 1100 °C is to discover most of the highly altered and disappeared mineral phases in the obtained XRD pattern before roasting by carrying out recrystallization for them. In addition, to show most of the occurred individual mineral bearing phases for $TiO_2$ other than psr/lpsr, and finally making some mineral phase conversions in solid state, for some associated mineral phases which may reflect and ensure the presence of some individual mineral phases for the same chemical component, especially $TiO_2$.

The sample of the non-roasted brown-coloured grains gave the composition of pseudorutile and rutile (Figure 6a) while it gave the composition of rutile, pseudobrookite, and quartz after roasting (Figure 6b).

The results ensure the presence of an individual $TiO_2$ phase (rutile), in association with the psr/lpsr alteration phases. In addition, it is obvious that the hexagonal psr structure is unstable in comparison with the tetragonal rutile structure where a considerable $TiO_2$ content escaped from the broken psr structure and diffused inside the rutile structure. Both of the remaining $TiO_2$ and most of the $Fe_2O_3$ contents of the collapsed psr are modified into the orthorhombic pseudobrookite structure which seems to be more stable under the new prevailing conditions.

The sample of the non-roasted black-coloured variety gave the composition of rutile, hematite, and quartz (Figure 6c) while it gave the composition of pseudobrookite, rutile, hematite, and quartz after roasting (Figure 6d). It is obvious that the presence of individual phases of $TiO_2$, $Fe_2O_3$, and $SiO_2$ affect the detection of individual psr/lpsr phases in an X-ray diffractogram pattern.

It was noticed that after roasting most of the stained and/or coated materials changed into translucent–transparent yellow and red primary rutile. In addition, some of fibrous and very-fine fragments of colourless silica are noticed in the container in which the magnetic grains are roasted.

In addition, a small representative sample (200 grains) was obtained from the relatively lighter grains (sp. Gr. < 4 $g/cm^3$), separated as a magnetic fraction at 1 A. The grains were split into two equal samples, the first sample subjected to the XRD while the second was subjected to the XRD after roasting at 1100 °C for one hour.The first sample was composed of rutile with minor anatase and quartz (Figure 6e), while the roasted sample gave the pattern of only rutile (Figure 6f). The detected mineral patterns are in accordance with the ASTM card numbers 4-0551, for rutile;19-182, for pseudorutile; 9-182, for pseudobrookite; 13-534, for hematite and 4-0477 for anatase.

The presence of anatase is not surprising. In some of the partially leucoxenated or partially replaced ilmenite parts (e.g., by sphene), the final obtained $TiO_2$ polymorph of these parts may differ to that due to the leucoxenation of the non-altered; or the non-replaced, ilmenite component of the same grain. In addition, the nature of the non-altered ilmenite component, whether normal ilmenite or ferriilmenite, the presence of exsolved titanhematite inside it, the orientation, and relative percentage of such exsolved minerallamellae in the host ilmenite may play another role in determining the formed type of the produced $TiO_2$ polymorph due to the difference in alteration rates because the rate of alteration could determine the type of the produced $TiO_2$ polymorph [13,37]. Thus, the altered grains can contain more than one type of $TiO_2$ polymorph due to its complete leucoxenation.

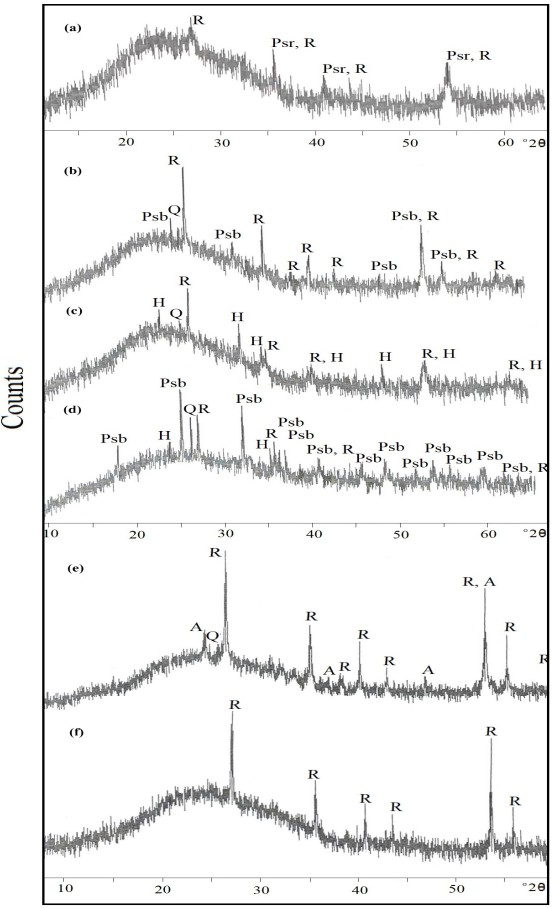

**Figure 6.** The X-ray diffractograms of the different altered ilmenite grains: the magnetic, brown-altered ilmenite of 0.5 ampere value before roasting (**a**) and after roasting (**b**); the magnetic black-altered ilmenite of 0.5 ampere value before roasting (**c**) and after roasting (**d**); the non-magnetic altered ilmenite separated as light Clerici's solution of 1ampere value before roasting (**e**) and after roasting (**f**).

### 3.4. The Role of $SiO_2$ and $Al_2O_3$ in Ilmenite Alteration Process

In many previous papers, both of the recorded $SiO_2$ and $Al_2O_3$ contents in the analyzed altered ilmenite spots are considered as impurities where they must be removed from the total sum of oxides followed by the recalculation and correction of the other analyzed oxides. However, in the case of lpsr phase spots, the most abundant analyzed oxide is $TiO_2$%,the spot which will be highly affected by the recalculation and correction process. It increases with a relatively greater percent than the other analyzed oxides, especially $Fe_2O_3$, the second major oxide of lpsr spot analysis. Hence, if $SiO_2$ and/or $Al_2O_3$ are not impurities in a definite analyzed spot, but one or both of them are originally associated with either $TiO_2$ or $Fe_2O_3$, then some misleading results are obtained due to the recalculation and corrections.

According to [24,38], aluminum and silicon are enriched due to iron depletion during the first stage of alteration, but their concentrations remain quite low: $Al_2O_3 \leq 0.4\%$ and $SiO_2 \leq 0.1\%$. For Ti/(Ti + Fe) > 0.7, the aluminum and silicon levels increase rapidly with increasing Ti/(Ti + Fe) ratios, to maximum values near 1.5% $Al_2O_3$ and 0.5% $SiO_2$. This increase is due to co-precipitation or adsorption of these elements from the surrounding soil solutions onto the freshly formed alteration products [24].

The occurrence of gibbsite and clay minerals within the pores of the weathered grains was postulated. Then, the increase in Al and Si contents with the increase in Ti/(Ti + Fe) ratio was a consequence of the increase in abundance of pores available for the

crystallization of clay minerals, as ilmenite and pseudorutile alter isovolumetrically to porous rutile [18].

The presence of Al and Si is either inherited from original inclusions, kaolinite alteration of these inclusions, or precipitated under pedogenic or early diagenetic conditions [37]. They are secondary mechanical enrichment [25].

In the process of ilmenite alteration and the formation of the various lpsr phases, it was detected that as both of the content of $TiO_2$ (within a definite content of $TiO_2$ ranges between 80% and 85%), and the associated str water increase, the associated contents of $Al_2O_3$ and $SiO_2$ increase. In fact, most of the detected lpsr spots have an appreciable content of mol water and contain relatively higher contents of $SiO_2$ and $Al_2O_3$. This may indicate the ability of these two oxides for bearing mol water or $OH^-$ anions. Thus, both $SiO_2$ and $Al_2O_3$ may be associated with the contained mol water in the altered grains [32].

Therefore, as the contents of str and/or mol water contained within the lpsr phases decrease; where the original total sum of oxides (OT) is more than 98% and the contained $TiO_2$ is more than 85%, the contents of $SiO_2$ and $Al_2O_3$ are highly depleted.

However, it was explained that the quantity of silica in most naturally occurring waters at near 25 °C is probably not controlled by the solubility of quartz. The decomposition of other silicates such as feldspars and the precipitation of various clays may be more important in regulating the quantities of silica found in natural waters [39].

The Nature of $SiO_2$ and $Al_2O_3$ Contents within psr and lpsr Analyzed Spots

The majority of the analyzed psr and lpsr spots of the altered ilmenite grainscontain $TiO_2$percent ranges between 58 and 80% and have appreciable contents of $Al_2O_3$ and/or $SiO_2$%. Most of their contents range between 2 and 3%. Only28 spots out of 586 spot analyses have $SiO_2$ and $Al_2O_3$ contents relatively greater than 3%. In a few spots, the contents of CaO rarely reach considerable values (10.13%). However, in the majority of the analyzed spots, CaO, $Na_2O$, and $K_2O$ are all present but in negligible amounts. In the forementioned 28 spots, the $SiO_2$ ranges between 0.75 and 13.38%, and the $Al_2O_3$ ranges between 0.31 and 9.13%. The calculated Si cations range between 0.04 and 0.77 while the calculated Al cations range between 0.02 and 0.59. In many of these spots, a clear negative correlation is present between $Fe_2O_3$ content and $SiO_2$ + $Al_2O_3$ contents. In fact, in many of these individual spots, if the contents of the included $SiO_2$% + $Al_2O_3$% are added to the present content of $Fe_2O_3$%, the sum of these three oxide contents gives the postulated accepted percentage of $Fe_2O_3$as in accordance with the contained $TiO_2$% in the analyzed spot, considering also that the detected spot is psr.

The investigation of psr and lpsr spot analyses reflects that the majority of spots containing a definite amount of str and may also be mol water contents. The deficiency of more than 100% for the sum of total analyzed oxides (OT) of each detected spot is considered corresponding to one or both of the two types of water. However, after applying the adopted psr Excel program, the decrease in the sum of total analyzed oxides (NT) below 100% will be corresponding only to the contained individual mol water.

It was detected that on applying the constructed Excel psr program for all the investigated psr/lpsr analyzed spots, the sum of the analyzed total oxides (NT), which includes the calculated str water, ranges between 94.1 and 107.7%. Only 376 spotsof psr/lpsr analyses have total oxide sum values less than 101%. The str and mol water ranges between 0.22 and 16.3%. The cationic iron ranges between 0.42 and 1.99. The recorded minimum total sum of oxides after applying the constructed Excel psr program equals 94.1% which ensures the presence of appreciable molecular water content in some of these analyzed spots.

For all the investigated psr and lpsr spots (586 analyses), it was considered that the analyzed five oxides $SiO_2$, $Al_2O_3$, CaO, $Na_2O$, and $K_2O$ are impurities as reported by almost all the previous published studies; thus, the aforementioned five oxides must be neglected and the percentages of the other remaining analyzed oxides must be recalculated. Then, the value of each analyzed oxide must bein the following correction factor: [100/100-($SiO_2$ + $Al_2O_3$ + CaO + $Na_2O$ + $K_2O$)]. On carrying this analysis out, it was no-

ticed that the calculated mol and/or str water ranges between 0.35 and 17%. After applying the constructed Excel psr program, the sum of the analyzed total oxides (NT), which includesthe calculated str water, ranges between 93.3 and 114.5%. Only 302 spots in thepsr analyses have total oxides sum values less than 101%. The $TiO_2$ contents after such correction for the majority of the analyzed spots range between 58 and 83.7%. Only seven spot analyses have corrected $TiO_2$ contents ranging between 84 and 93.6%. The cationic iron ranges between 0.12 and 1.97.

However, after the careful investigation of the analyzed psr/lpsr spots, the following assumptions about the presence of such detected impurity oxides can be summarized:

1. Both $SiO_2$ and $Al_2O_3$, the two major impurity oxides of the analyzed five impurity oxides, can be present as individual oxides inside the fissures and cracks of the altered ilmenite grains. Then, by neglecting them and recalculation the percentages of the remaining oxides of each spot, the spots should give an accepted result upon applying the constructed Excel psr program. This case was not achieved with the majority of the analyzed spots in this study.

2. One or more of these impurity oxides are associated with some or all of the analyzed $Fe_2O_3$ in addition to others which may be remnants due to alteration of a preexisting definite silicate mineral containing minor amounts of $TiO_2$. At definite geological conditions, considerable amounts of $SiO_2$, $Al_2O_3$, CaO, and $Fe_2O_3$ present in the silicate mineral composition are removed and an enrichment of the contained $TiO_2$ content occurred. There are several spots of several grains which have such circumstances.

3. In other cases, $SiO_2$ and/or $Al_2O_3$ are a replacement for an individual phase of $Fe_2O_3$. In such cases, both $Fe_2O_3$ and $TiO_2$ are two individual phases after ilmenite or follow the breakdown of psr or lpsr in the final alteration stages of ilmenite. Geothites and hematites can incorporate quite large amounts of Al in their structures [40].

4. The fourth assumption is that, in several other cases, both $SiO_2$ and $Al_2O_3$ are not present as impurities, they play an important role in the late stages of ilmenite alteration. They act as donors or acceptors of $H_2O/OH^-$ in the alteration mechanisms of lpsr. In fact, the Al-OH bond is greater than the Fe-OH bond [41]. However, the enrichments of these oxides are noticed in the alteration mechanisms in which either the oxygen is replaced with $OH^-$ or during the enrichment of the individual $TiO_2$ phase during the losing of most of the str and/or mol water from the altered lpsr formula structure. Several authors studied the conversion of $Fe(OH)_3$ gel to β-FeOOH and α-$Fe_2O_3$ in the presence and absence of silicate ions [42,43]. At 60 °C, the effect of silicate ions resembles that of phosphate, and completely suppresses these two conversions, while it is in contrast to the influence of sulphate ions which accelerate the conversion of the $Fe(OH)_3$ gel to β-FeOOH and delay the conversion from β-FeOOH to α-$Fe_2O_3$ [43]. Then, $SiO_2$ and/or $Al_2O_3$ may play a definite roles in some conversions of Fe- or Ti-oxyhydroxides of some suggested alteration scenarios by the present author (under publication).

In some spot analyses of altered ilmenite grains, $TiO_2$ ranges between 80 and 90%; 19 spot analyses are detected to contain relatively higher contents of $SiO_2$, $Al_2O_3$, and CaO. The $SiO_2$ ranges between 1.78 and 11.84% and the $Al_2O_3$ ranges between 0.09 and 5.71%, while CaO ranges between 0.1 and 6.76%. All of these spot analyses have accepted sum total oxides very close to 100%, before and after neglecting these three oxides and correcting the percentages of the remaining analyzed oxides. It is obvious that these three oxides are impurities and neither contain mol nor str water inside their structures. In these cases, there is not a psr formula structure and most of the contained $TiO_2$ and $Fe_2O_3$ are individual phases or both of them are present as one phase other than psr, most probably ferriferous rutile. In fact, on applying the constructed psr Excel program for these spot analyses, which do not have apsr structure, before and after neglecting the impurity oxides, the obtained results are not accepted. The sum of the total analyzed oxides in addition to the calculated structural water ranges between 105.06 and 112.35% before neglecting the oxides and ranges between 108.53 and 117.3% after neglecting them and recalculating the

values forthe other remaining oxides.Then, such results are obtained when there is nopsr formula structure or at least not all the present major oxides in the analyzed spot (i.e., $TiO_2$) are present only with psr. There are also other individual phases containing some of this major oxide ($TiO_2$).

According to the last explanation, the relatively enriched contents of $SiO_2$ and/or $Al_2O_3$ in some secondary rutile grains can be explained as most of the $SiO_2$% is associated with mol water or bearing for mol and/or str water necessary for the leachability of $Fe^{3+}$ from the psr structure. It will be settled and remains inside the interspaces and pores of the remaining lpsr phase structure. During the growth of the final micro-, crypto-, or the triple-twinned arrangement of secondary rutile polymorph starting from the core to the outside of the highly altered grain, the $SiO_2$ molecules may be either leached out from the interspaces or concentrated toward the outside on the periphery of the obtained final alteration products.

## 4. Conclusions

It has been found that the contents of $TiO_2$ and $Fe_2O_3$ for the investigated lpsr spots are in the range of 59.16–86.56% and 37.3–6.68%, respectively. The Ti/(Ti + Fe) ratio ranges between 0.60 and 0.88 and the well-defined accepted psr/lpsr chemical formulas are as follows: $Fe_{2.01-0.50}Ti_3O_{8.97-4.5}(OH)_{0.03-4.5}$. Most of the detected lpsr phases have an extended composition range in the most stable psr phase, $Fe_2Ti_3O_9$. Finally, the detected lpsr molecular formula of the lowest cationic iron content (0.5) has been recorded in the present study.

It has been found that in most of the investigated lpsr spots, the mechanism of ilmenite alteration may be changed in the region of 68–70% $TiO_2$ where neither all the analyzed $TiO_2$ is included in the lpsr phase nor the calculated value of structural water within the lpsr molecular formula is correct. There are other individual phases containing some of $TiO_2$ or Ti-oxyhydroxide. These individual phases are most probably separated from the broken lpsr phase. Then, at a definite lpsr phase formula, the structure will be gradually broken up. At this moment, the lpsr structure not only loses some of its Ti content but also some of its structural water will be lost with Ti.

The present study ensures that both $SiO_2$ and/or $Al_2O_3$ may play very important roles in the alteration processes of ilmenite, especially in the stages of formation of psr/lpsr and secondary rutile. Researchers must reconsider the role of $SiO_2$ and $Al_2O_3$ in this type of alteration which may be in other alteration types. In fact, in many of the investigated lpsr phases, both $SiO_2$ and $Al_2O_3$ contents are not present in the analyzed spots as impurities. These two oxides may play an important role in the late stages of ilmenite alteration. They may act as donors or acceptors of $H_2O/OH^-$ in the alteration mechanisms of lpsr. However, the enrichments of these two oxides are noticed in the alteration mechanisms in which either the oxygen is replaced with $OH^-$ or during the enrichment of the individual $TiO_2$ phase during the losing of most of the str and/or mol water from the collapsed lpsr formula structure. Both $SiO_2$ and/or $Al_2O_3$ may play definite roles in some conversions of Fe- or Ti-oxyhydroxide individual phases after the collapse of the preexisting lpsr phases and the formation of ferriferous rutile or secondary rutile.

It was detected that the investigated altered ilmenite spots having the lowest iron contents (0.49–0.42) are composed mainly of individual $TiO_2$ and $Fe_2O_3$ phases due to collapse of the majority of the preexisting psr/lpsr phases. Then, the cationic Fe contents ranging between 0.42 and 0.49 were discarded as minimum values for the existence of psr/lpsr phases. Then, the collapse of psr structure seems to be at cationic iron content just below 0.5.

Not only is the use of the XRD technique for the identification of existed mineral phases, but it is also for detection of some mineral alteration mechanisms by making some reactions and conversions for prediction. However, the XRD results ensure the presence of the individual $TiO_2$ phase (rutile), in association with the psr/lpsr alteration phases. Additionally, on roasting the altered grains at 1100 °C, it is obvious that the hexagonal psr

structure is unstable in comparison with the tetragonal rutile structure, a considerable $TiO_2$ content escaped from the broken psr structure and diffused inside the rutile structure.

Moreover, some of the fibrous and very fine fragments of colourless silica are noticed in the container in which the magnetic grains are roasted. This result ensures that the presence of silica, and possibly alumina, is not an impurity, otherwise it can remain with the formed rutile or pseudobrookite.

The altered ilmenite grains, separated individually at 0.5 and 1 ampere values, have an enrichment of various mineral types including altered silicate minerals, silica, sphene, and individual mineral phases of Ti and Fe. Furthermore, the grains have high contents of str water and considerable volumes of voids and cracks. All of these different components are the reason for the relatively lower magnetic characters of the investigated altered ilmenite grains.

**Supplementary Materials:** The following supporting information can be downloaded at: https://www.mdpi.com/article/10.3390/geosciences13060170/s1, File S1: The calculation of the molecular formula of ilmenite for only one spot analysis; File S2: The calculation of the molecular formula of ilmenite for several spot analyses; File S3: The calculation of the molecular formula of leached ilmenite for only one spot analysis; File S4: The calculation of the molecular formula of leached ilmenite for several spot analyses; File S5: The calculation of the molecular formula of pseudorutile/leached pseudorutile for only one spot analysis; File S6: The calculation of the molecular formula of the pseudorutile/leached pseudorutile for several spot analyses.

**Funding:** This research received no external funding.

**Data Availability Statement:** Not applicable.

**Acknowledgments:** The author wishes to express his deepest gratitude to H.J. Massonne, Thomas Theye, and the microprobe unit of the Institute of Mineralogy and Crystal Chemistry, Stuttgart University, Germany, for providing the microprobe analytical facilities.

**Conflicts of Interest:** The author declares no conflict of interest.

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
