# Peer review of "Study of the Mineralogical and Chemical Compositions of the Weakly Magnetic Fractions of the Egyptian Black Sand Altered Ilmenite"

_geosciences, doi:10.3390/geosciences13060170_

Round 1
Reviewer 1 Report
The manuscript "The Alteration Processes of the Weakly Magnetic Egyptian Black Sand Altered Ilmenite" by Mohamed I Moustafa is devoted to the study of Egyptian Black Sand ilmenite and the processes of its alteration. The paper presents analytical data on the chemical composition of various types of ilmenite, on the basis of which the formula units of the detected phases were calculated. The microprobe chemical analyzes of the studied grains are given in the tables. For a better understanding of these data, in the table opposite the serial number of the analysis point, it is necessary to give the name of the mineral phase. The text is replete with data on variations in the chemical composition of grains, which essentially repeats the information from the tables. The manuscript has a section called The XRD results. At the same time, in the Materials and Methods section, there is no information at what parameters and on what equipment these results were obtained. It is also not clear for what purpose some grains were heated at 1100 oC. The X-ray diffractograms in Figure 6 need to be presented in the best possible quality. In general, the presented work contains interesting data and conclusions on the processes of alteration of Egyptian Black Sand ilmenite.
Author Response
Dear Reviewer
The manuscript was revised according to your valuable comments. I hope that I have been able to fulfill most of the requirements in revising the manuscript. The revision was as follows:
Comments and Suggestions for Authors:
- The manuscript "The Alteration Processes of the Weakly Magnetic Egyptian Black Sand Altered Ilmenite" by Mohamed I Moustafa is devoted to the study of Egyptian Black Sand ilmenite and the processes of its alteration. The paper presents analytical data on the chemical composition of various types of ilmenite, on the basis of which the formula units of the detected phases were calculated. The microprobe chemical analyzes of the studied grains are given in the tables. For a better understanding of these data, in the table opposite the serial number of the analysis point, it is necessary to give the name of the mineral phase.
A new Column was inserted in each table and titled as “Mineral phase” were the corresponding mineral phases of each spot analysis is given.
Also, according to Reviewer 1:
- The text is replete with data on variations in the chemical composition of grains, which essentially repeats the information from the tables.
Note that not all the analyzed spots were included in the Tables. The size of Tables was reduced to the half. Then both of the minimum, maximum and average values for all the analyses of a definite grain should be remembered. However some of these repeats will be taken into consideration during the revision of the article.
According to Reviewer 1:
- At the same time, in the Materials and Methods section, there is no information at what parameters and on what equipment these results were obtained.
- Also, item 4 in the given table of Reviewer’1 Report: Are the methods adequately described? “can be improved”.
Taking Materials and Methods into consideration, there was another first article in addition to the current article. It was dealing with the study of the strongly magnetic altered ilmenite fractions (separated at 0.1-0.35 ampere values). In this first article, both of the used microprobe analytical conditions, the methods of calculations of all the constructed Excel softwares necessary for calculating the molecular formulas of ilmenite, leached ilmenite, psr & lpsr and finally the different mineral processing stages to obtain the various strongly to weakly magnetic altered ilmenite fractions. The first article was sent also, to the same current Journal. It is not accepted. Now this first article in addition to another third article are published in another Journal. They are in the reviewing stage since about two weeks. I have no problem to attach the essence and algorithm of calculations as Supplementary Materials to this current article. But I do not know how legal that is. In fact, in the first article, all items concerned Materials and Methods are included in the main text of the article, not as supplementary materials. They include 7 A4 pages.
However, under the item 2.Materials and Methods, the Lines from 100-118, Page 3 are replaced with the following paragraphs:
“Using the difference in physical characters between the various economic minerals [5, 28, 29, 31, 32], the collected surfacial naturally highly concentrated beach raw sands were processed using the following equipments:
-The Reading cross–belts magnetic separator for magnetic separation where the raw sample is differentiated into three fractions, a ferromagnetic fraction, a bulk magnetic fraction and a bulk non magnetic fraction.
-The Full size Wilfley shaking tables for wet-gravity concentration of the obtained bulk non magnetic fraction.
-The Carpco (HP 167) high tension roll-type electrostatic separator for treating the obtained tabled concentrate to obtain a bulk rutile conductor fraction and a bulk zircon non conductor fraction.
-The Carpco (MIH 13-231-100) industrial high intensity induced roll dry magnetic separator for magnetic separation of the obtained bulk rutile conductor fraction.
-The Frantz isodynamic magnetic separator. The obtained three successive magnetic fracions of the last treatment stage are mixed together as a bulk magnetic fraction. It is composed of hematite, ilmeno-hematite, different varieties of magnetic primary rutile [32], and various grades of altered ilmenite grains, in addition also to minor of Cr-bearing minerals and other magnetic minerals [5]. A relatively smaller representative sample is obtained from the bulk magnetic fraction and subjected to magnetic differentiation using the Frantz isodynamic magnetic separator where the used adjustment of operating conditions are longitudinal slope of 20o, side slope of 5o, feeding rate of 30 g/hour and different successive ampere values; 0.1, 0.2, 0.25. 0.35, 0.5 and 1 ampere where six magnetic fractions and only one non magnetic fraction are obtained. The altered ilmenite grains obtained in the two individual magnetic fractions separated at 0.5 and 1 ampere values which are investigated in this article.
-The microscopic investigation-the altered ilmenite varieties obtained of these separated two individual magnetic fractions are investigated using the binocular and reflected microscopes.
-The microprobe analysis-the investigation of the different altered ilmenite grains were carried out by a Cameca SX-100 electron microprobe analyzer (EMPA); institute of mineralogy and crystal chemistry, Stuttgart University, Germany. The microprobe instrument is equipped with three wavelength dispersive spectrometers (WDS) and an energy dispersive spectrometer (EDS). The whole surface of the polished sections was examined by back scattered electron (BSE) images, so that the grain with e.g. 10 µm size; or even smaller could be detected. The analytical conditions were 15 kV accelerating voltage; 15 nA electron current; 180s counting time for each analyzed spot in theinvestigated grains and a focused electron beam diameter of 1 to 4 µm. The following standards were used : diopside for Mg and Ca, albite for Na, Corundum for Al, orthoclase for Si and K, rutile for Ti, rhodonite for Mn, Fe2O3 for Fe, Cr2O3 for Cr , V for V and sphalerite for Zn. Lines used for analysis were Kα for each of the analyzed elements. For each detected altered ilmenite variety, a definite number of grains are picked individually and polished for the investigation using the microprobe.
-The X-ray diffraction instrument (XRD). Philips X-ray generator (PW 3710/31) with automatic sample changer (PW 1775; 21 position) using scintillation counter, Cu-target tube and Ni filter at 40 kV and 30 mA was used. This instrument is connected with computer system using X-40 diffraction program and ASTM cards for mineral identification”.
After the last modified paragraph, another new paragraph is added:
“However, all the used constructed Excel softwares for the calculation of different molecular formulas of mineral phases; which is the final result of the essence and algorithm of calculations, are given in the article as supplementary materials (S1-S6)”.
Also, according to Reviewer 1:
- It is also not clear for what purpose some grains were heated at 1100 o The X-ray diffractograms in Figure 6 need to be presented in the best possible quality. In general, the presented work contains interesting data and conclusions on the processes of alteration of Egyptian Black Sand ilmenite.
Under the title 3.3. The XRD Results, after the first paragraph the following paragraph is included in Lines from 627 to 633, Page 23.:
“The purpose of roasting the altered grains at 1100 oC is to discover most of the highly altered and disappeared mineral phases in the obtained XRD pattern before roasting by making recrystallization for them. Also, to show most of the occurred individual mineral phases bearing for TiO2 other than psr/lpsr, and finally making some mineral phases conversion in solid state, for some associated mineral phases which may reflect and ensure the presence of some individual mineral phases for the same chemical component especially TiO2”.
Also, Fig.6, Page 26 was revised.
According to Reviewer 1, the following table was given:
|
Yes |
Can be improved |
Must be improved |
Not applicable |
|
|
Does the introduction provide sufficient background and include all relevant references? |
( ) |
(x) |
( ) |
( ) |
|
Are all the cited references relevant to the research? |
( ) |
(x) |
( ) |
( ) |
|
Is the research design appropriate? |
( ) |
(x) |
( ) |
( ) |
|
Are the methods adequately described? |
( ) |
( ) |
(x) |
( ) |
|
Are the results clearly presented? |
( ) |
(x) |
( ) |
( ) |
|
Are the conclusions supported by the results? |
( ) |
(x) |
( ) |
( ) |
- Introduction:
Under the item 1.Introduction, in Lines 64, 65: the paragraph “……….was reported by [6], and which was considered as merely weathered ilmenite by [7]”.
Is rewritten as “ ……. was reported by [6]. It was considered as merely weathered ilmenite by [7]”.
In Line 74, the word “However” is written before [22]
However, [22] does not agree with the explanation of [21], unless the primary pseudobrookite was early present in the source area.
In Line 76, “For ilmenite alteration of Ti/(Ti+Fe)> 0.7,”
Is rewritten as “In case of ilmenite alteration has Ti/(Ti+Fe)> 0.7,”
Also, the Lines from 85-95, Page 3 are reformulated as follows:
In studying a sample from Rosetta ilmenite concentrate of Egyptian black sand, it was explained that although the alteration to psr is observed, further alteration to leucoxene is very rare [27]. Most of the mineralogical features for both of the homogeneous Egyptian black sand ilmenite, the different exsolved intergrowths between ilmenite-other mineral components and the partially altered ilmenite were explained [28, 29]. In these two last studies, it was noticed that the presence of molecular water is very important in the alteration process of ilmenite or some associated silicate mineral impurities. Also, the chemical composition of the highly altered leucoxenated Egyptian beach ilmenite grains reflects that the TiO2 ranges between 59.45 and 89.72%, the total iron content (Fe2O3) varies from 2.34 to 32.68%, whereas the SiO2 content varies from 0.89 to 8.19% [30].
It is obvious that in the last reformulated lines, the number of some references were changed: [29] to [27]; and [27, 28] to [28,29].
The two Lines 103, 104, Page 3 are reformulated in the Lines from 104-112 as follows:
In the present article, some of the weakly magnetic altered ilmenite grains are investigated to explain their mineralogical and chemical composition characters. The purpose of the work is to detect both of the different lpsr phases, the lowest iron content in these phases and the most stable molecular formula of the detected lowest lpsr phase. Also, the prediction of the role of SiO2 and Al2O3 contents in ilmenite alteration. Are they just impuritiesas reported in all previous studies?, or they play a definite role in ilmenite alteration stages. In fact, if these targets are explained then, both of the most accepted formed lpsr molecular formula, real role of SiO2&Al2O3 contents in minerals alteration and illustration of some given previous psr/lpsr molecular formulas can be achieved.
The Lines from 93-96 are deleted:
“The different alteration phases and their corresponding molecular formulas limits will be explained. The most accepted mechanism of their alteration processes are explained. The role of both of Al2O3 and SiO2 in the process of ilmenite alteration will be also investigated”.
Now, according to the Reviewer, “Does the introduction provide sufficient background and include all relevant references? Can be improved”.
I think that the introduction was improved.
- The cited References:
I think that all of the cited References are relevant to the research.
- Is the research design appropriate? “can be improved”.
- Are the results clearly presented? “can be improved”.
In fact, many of words and several Lines in the text were removed or substituted by others. All of these changes are clearly shown and marked up using the “Track Changes” function such that any changes canbe easily viewed by the editors and reviewers.
In Line 193, the word obtained was replaced by magnetic
In Line 199, the sentence “Others are coated or stained with silica” is reformulated to “Other grains are stained or coated with silica”.
The sentence of Lines 201-203 is reformulated to be “Taking the brown coloured grains of the magnetic fraction into consideration, three groups of grains are investigated as follows:”
Line 204 was modified.
Line 205, was modified.
Line 207, was modified.
The lines 208, 212, 201, 214 were modified.
The paragraph in Lines from 215-218 is modified.
Some modifications were carried out in the Lines 224, 228, 230, 233, 236, 237-242, 245, 248-251.
Some modifications in the Lines 254, 258, 260, 261, 263, 264, 265, 267, 268, 273, 274.
A sentence was added in Line 280.
The sentence in Lines 284-286 was reformulated.
Some modifications in Lines 289, 291, 292, 293, Page 8.
Some modifications in Lines 314-321, 323, 324, 330, 335, 336, 342-344, 346, Page 9.
Some modifications in Lines 351-353, 357, 358, 365, 371, 374, 378, 380, 383, Page 10.
Some modifications in Lines 384-387, 391, 393, Page 11.
Some modifications in Lines 433, 435, 440-444, the paragraph in Lines 445-455, Line 456, 461, 469, Page 16.
Some modifications in Lines 470-472, 474, Page 17.
Some modifications in Lines 517-520, 523, 524, 526, 527, 535, 536, Page 20.
Some modifications in Lines 551, 573, 574, 579, Page 21.
Some modifications in Lines 580-581, 586-589, 594, 596-599, 605, 608-611, Page 22.
The paragraph of lines 624-626 became in Lines 634-636, Page 23.
Some modifications in Line 639, Page 23
Some modifications in Lines 659, 660, 662-664, 673, Page 25.
Some modifications in Lines 698, 699, 705-707, Page 26.
Some modifications in Lines 709, 711, 724, 725, Page 27.
Some modifications in Lines 751-753, 758, 759, 762, 764, 773, 775, 776, 778, 782-784, Page 28.
Some modifications in Lines 792, 799, 809, Page 29.
Some modifications in Lines 830-832,841, Page 30.
- Are the conclusions supported by the results? Can be improved.
Almost all conclusions are reformulated to be as follows:
It was concluded that the contents of TiO2 and Fe2O3 of the investigated lpsr spots range between 59.16-86.56%, and 37.3-6.68%, respectively. The Ti/(Ti+Fe) ratio range between 0.60 and 0.88 and the well defined accepted psr-lpsr chemical formulas are as follows: Fe2.01-0.50Ti3O8.97-4.5(OH)0.03-4.5. Most of the detected lpsr phases have extended composition range of the most stable psr phase, Fe2Ti3O9. Finally, the detected lpsr molecular formula of the lowest cationic iron content (0.5) is recorded in the present study.
It was concluded that in most of investigated lpsr spots, the mechanism of ilmenite alteration may be changed in the region of 68-70% TiO2 where neither all the analyzed TiO2 is included in the lpsr phase nor the calculated value of structural water within the lpsr molecular formula is correct. There are other individual phases containing some of TiO2 or Ti-oxyhydroxide. These individual phases are most probably separated from the broken lpsr phase.
Also, in many of the investigated lpsr phases, both of SiO2 and Al2O3 contents are not present in the analyzed spots as impurities. These two oxides may play an important role in the late stages of ilmenite alteration. They may act as doners or acceptors of H2O/OH- in the alteration mechanisms of lpsr. However, the enrichments of these two oxides are noticed in the alteration mechanisms in which either the oxygen is replaced with OH- or during the enrichment of the individual TiO2 phase during the losing of most of str and/or mol water from the collapsed lpsr formula structure. Both of SiO2 and/or Al2O3 may play a definite roles in some conversions of Fe- or Ti- oxyhydroxides individual phases after the collapse of the preexisting lpsr phases and the formation of forriferous rutile or secondary rutile.
It was detected that the investigated altered ilmenite spots having the lowest iron contents (0.49-0.42) are composed mainly of individual TiO2 and Fe2O3 phases due to collapse of the majority of the preexisting psr/lpsr phases. Then, The cationic Fe contents ranges between 0.42 and 0.49 are discarded as minimum values for the existence of psr-lpsr phases. Then, the collapse of psr structure seems to be at cationic iron content just below 0.5.
The XRD results ensure the presence of individual TiO2 phase (rutile), in association with the psr/lpsr alteration phases. Also, on roasting the altered grains at 1100 oC, it is obvious that the hexagonal psr structure is unstable in comparison with the tetragonal rutile structure, a considerable TiO2 content escaped from the broken psr structure and diffused inside the rutile structure.
Also, some of fibrous and very fine fragments of colourless silica are noticed in the container in which the magnetic grains are roasted. This result ensures that the presence of silica; and may be alumina, are not as impurities otherwise it can remains also with the formed rutile or pseudobrookite.
The altered ilmenite grains separated individually at 0.5 and 1 ampere values, have an enrichment of various mineral types including altered silicate minerals, silica, sphene, and individual mineral phases of Ti and Fe. Also, the grains have high contents of str water and considerable volumes of voids and cracks. All of these different components are the reason of the relatively lower magnetic characters of the investigated altered ilmenite grains.
With my best regards
M.I.Moustafa

Reviewer 2 Report
The article is devoted to the study of the mineralogical composition of the low-magnetic fraction of the Egyptian Black Sand Beach Ilmenite. I can recommend this article for publication after making additional clarifications and corrections. Notes:
1. The title of the article refers to the processes of alteration of ilmenite. However, the author does not study processes, but its results. The author himself indicates that some of the altered ilmenite are investigated to explain their mineralogical and chemical composition characters. Therefore, it is obvious that the title does not match the content. The title should indicate that the mineralogical and chemical composition of some of the altered ilmenite is being studied.
2. The statement of the research problem and the purpose of the article are not clear. The task statement is as follows:
"In recent two papers, most of the mineralogical features of the homogeneous Egyptian black sand ilmenite, the different exsolved intergrowths between ilmenite-other mineral components and the partially altered ilmenite grains were explained [27,28]. Although the alteration to pseudorutile is observed , further alteration to leucoxene is very rare in the studied sample from Rosetta ilmenite concentrate [29]."
References [27-29] belong to the author. Thus, he sets himself the task of investigating the mineralogical composition of the low-magnetic fraction of ilmenite. For what? What is the relevance of such research? What is the purpose, i.e. what is their practical value? The description of the research problem statement needs to be improved.
It is also necessary to clearly state the purpose of the research. To do this, use the words: "the purpose of the work is ...". It should be noted that the study cannot be the purpose of the article, as this is a way to achieve its goal.
3. The methodology for calculating the chemical compositions of the studied objects is not presented. It is stated that "the same constructed Excel softwares of [31], to calculate the molecular formulas of leached ilmenite, pseudorutile, and leached pseudorutile are used". However, the publication [31] is still unavailable. I believe that also in this article it is necessary to describe the essence of such calculations. The essence and algorithm of calculations should be clear to the reader.
Author Response
Dear Reviewer
The manuscript was revised according to your valuable comments. I hope that I have been able to fulfill most of the requirements in revising the manuscript. The revision was as follows:
Comments and Suggestions for Authors
The article is devoted to the study of the mineralogical composition of the low-magnetic fraction of the Egyptian Black Sand Beach Ilmenite. I can recommend this article for publication after making additional clarifications and corrections.
Notes:
- The title of the article refers to the processes of alteration of ilmenite. However, the author does not study processes, but its results. The author himself indicates that some of the altered ilmenite are investigated to explain their mineralogical and chemical composition characters. Therefore, it is obvious that the title does not match the content. The title should indicate that the mineralogical and chemical composition of some of the altered ilmenite is being studied.
The title was changed to “study of the Mineralogical and Chemical Compositions of the Weakly Magnetic Fractions of the Egyptian Black Sand Altered Ilmenite”.
- The statement of the research problem and the purpose of the article are not clear. The task statement is as follows:
"In recent two papers, most of the mineralogical features of the homogeneous Egyptian black sand ilmenite, the different exsolved intergrowths between ilmenite-other mineral components and the partially altered ilmenite grains were explained [27,28]. Although the alteration to pseudorutile is observed , further alteration to leucoxene is very rare in the studied sample from Rosetta ilmenite concentrate [29]."
References [27-29] belong to the author. Thus, he sets himself the task of investigating the mineralogical composition of the low-magnetic fraction of ilmenite. For what? What is the relevance of such research? What is the purpose, i.e. what is their practical value? The description of the research problem statement needs to be improved.
It is also necessary to clearly state the purpose of the research. To do this, use the words: "the purpose of the work is ...". It should be noted that the study cannot be the purpose of the article, as this is a way to achieve its goal.
Under the item 1.Introduction, in Lines 64, 65: the paragraph “……….was reported by [6], and which was considered as merely weathered ilmenite by [7]”.
Is rewritten as “ ……. was reported by [6]. It was considered as merely weathered ilmenite by [7]”.
In Line 74, the word “However” is written before [22]
However, [22] does not agree with the explanation of [21], unless the primary pseudobrookite was early present in the source area.
In Line 76, “For ilmenite alteration of Ti/(Ti+Fe)> 0.7,”
Is rewritten as “In case of ilmenite alteration has Ti/(Ti+Fe)> 0.7,”
Also, the Lines from 85-95, Page 3 are reformulated as follows:
In studying a sample from Rosetta ilmenite concentrate of Egyptian black sand, it was explained that although the alteration to psr is observed, further alteration to leucoxene is very rare [27]. Most of the mineralogical features for both of the homogeneous Egyptian black sand ilmenite, the different exsolved intergrowths between ilmenite-other mineral components and the partially altered ilmenite were explained [28, 29]. In these two last studies, it was noticed that the presence of molecular water is very important in the alteration process of ilmenite or some associated silicate mineral impurities. Also, the chemical composition of the highly altered leucoxenated Egyptian beach ilmenite grains reflects that the TiO2 ranges between 59.45 and 89.72%, the total iron content (Fe2O3) varies from 2.34 to 32.68%, whereas the SiO2 content varies from 0.89 to 8.19% [30].
It is obvious that in the last reformulated lines, the number of some references were changed: [29] to [27]; and [27, 28] to [28,29].
The two Lines 103, 104, Page 3 are reformulated in the Lines from 104-112 as follows:
In the present article, some of the weakly magnetic altered ilmenite grains are investigated to explain their mineralogical and chemical composition characters. The purpose of the work is to detect both of the different lpsr phases, the lowest iron content in these phases and the most stable molecular formula of the detected lowest lpsr phase. Also, the prediction of the role of SiO2 and Al2O3 contents in ilmenite alteration. Are they just impuritiesas reported in all previous studies?, or they play a definite role in ilmenite alteration stages. In fact, if these targets are explained then, both of the most accepted formed lpsr molecular formula, real role of SiO2&Al2O3 contents in minerals alteration and illustration of some given previous psr/lpsr molecular formulas can be achieved.
The Lines from 93-96 are deleted:
“The different alteration phases and their corresponding molecular formulas limits will be explained. The most accepted mechanism of their alteration processes are explained. The role of both of Al2O3 and SiO2 in the process of ilmenite alteration will be also investigated”.
Now, according to the Reviewer, “Does the introduction provide sufficient background and include all relevant references? Can be improved”.
I think that the introduction was improved.
- The methodology for calculating the chemical compositions of the studied objects is not presented. It is stated that "the same constructed Excel softwares of [31], to calculate the molecular formulas of leached ilmenite, pseudorutile, and leached pseudorutile are used". However, the publication [31] is still unavailable. I believe that also in this article it is necessary to describe the essence of such calculations. The essence and algorithm of calculations should be clear to the reader.
Taking Materials and Methods into consideration, there was another first article in addition to the current article. It was dealing with the study of the strongly magnetic altered ilmenite fractions (separated at 0.1-0.35 ampere values). In this first article, both of the used microprobe analytical conditions, the methods of calculations of all the constructed Excel softwares necessary for calculating the molecular formulas of ilmenite, leached ilmenite, psr & lpsr and finally the different mineral processing stages to obtain the various strongly to weakly magnetic altered ilmenite fractions. The first article was sent also, to the same current Journal. It is not accepted. Now this first article in addition to another third article are published in another Journal. They are in the reviewing stage since about two weeks. I have no problem to attach the essence and algorithm of calculations as Supplementary Materials to this current article. But I do not know how legal that is. In fact, in the first article, all items concerned Materials and Methods are included in the main text of the article, not as supplementary materials. They include 7 A4 pages.
However, under the item 2.Materials and Methods, the Lines from 100-118,Page 3 are replaced with the following paragraphs:
“Using the difference in physical characters between the various economic minerals [5, 28, 29, 31, 32], the collected surfacial naturally highly concentrated beach raw sands were processed using the following equipments:
-The Reading cross–belts magnetic separator for magnetic separation where the raw sample is differentiated into three fractions, a ferromagnetic fraction, a bulk magnetic fraction and a bulk non magnetic fraction.
-The Full size Wilfley shaking tables for wet-gravity concentration of the obtained bulk non magnetic fraction.
-The Carpco (HP 167) high tension roll-type electrostatic separator for treating the obtained tabled concentrate to obtain a bulk rutile conductor fraction and a bulk zircon non conductor fraction.
-The Carpco (MIH 13-231-100) industrial high intensity induced roll dry magnetic separator for magnetic separation of the obtained bulk rutile conductor fraction.
-The Frantz isodynamic magnetic separator. The obtained three successive magnetic fracions of the last treatment stage are mixed together as a bulk magnetic fraction. It is composed of hematite, ilmeno-hematite, different varieties of magnetic primary rutile [32], and various grades of altered ilmenite grains, in addition also to minor of Cr-bearing minerals and other magnetic minerals [5]. A relatively smaller representative sample is obtained from the bulk magnetic fraction and subjected to magnetic differentiation using the Frantz isodynamic magnetic separator where the used adjustment of operating conditions are longitudinal slope of 20o, side slope of 5o, feeding rate of 30 g/hour and different successive ampere values; 0.1, 0.2, 0.25. 0.35, 0.5 and 1 ampere where six magnetic fractions and only one non magnetic fraction are obtained. The altered ilmenite grains obtained in the two individual magnetic fractions separated at 0.5 and 1 ampere values which are investigated in this article.
-The microscopic investigation-the altered ilmenite varieties obtained of these separated two individual magnetic fractions are investigated using the binocular and reflected microscopes.
-The microprobe analysis-the investigation of the different altered ilmenite grains were carried out by a Cameca SX-100 electron microprobe analyzer (EMPA); institute of mineralogy and crystal chemistry, Stuttgart University, Germany. The microprobe instrument is equipped with three wavelength dispersive spectrometers (WDS) and an energy dispersive spectrometer (EDS). The whole surface of the polished sections was examined by back scattered electron (BSE) images, so that the grain with e.g. 10 µm size; or even smaller could be detected. The analytical conditions were 15 kV accelerating voltage; 15 nA electron current; 180s counting time for each analyzed spot in theinvestigated grains and a focused electron beam diameter of 1 to 4 µm. The following standards were used : diopside for Mg and Ca, albite for Na, Corundum for Al, orthoclase for Si and K, rutile for Ti, rhodonite for Mn, Fe2O3 for Fe, Cr2O3 for Cr , V for V and sphalerite for Zn. Lines used for analysis were Kα for each of the analyzed elements. For each detected altered ilmenite variety, a definite number of grains are picked individually and polished for the investigation using the microprobe.
-The X-ray diffraction instrument (XRD). Philips X-ray generator (PW 3710/31) with automatic sample changer (PW 1775; 21 position) using scintillation counter, Cu-target tube and Ni filter at 40 kV and 30 mA was used. This instrument is connected with computer system using X-40 diffraction program and ASTM cards for mineral identification”.
These new paragraphs include now the Lines from 106, Page 3 to 152, Page 4.
After the last modified paragraph, another new paragraph is added:
“However, all the used constructed Excel softwares for the calculation of different molecular formulas of mineral phases; which is the final result of the essence and algorithm of calculations, are given in the article as supplementary materials (S1-S6).”
With my best regards
M.I.Moustafa

Round 2
Reviewer 1 Report
In general, the author did a good job of finalizing the manuscript. The revision of the methodological part deserves special attention. However, there are a few more editorial comments.
1. The use of the "&" sign in the designation SiO2&Al2O3 and in other cases according to the text. I would recommend using the spelling of the word "and".
2. Design of tables.
The top line of the table, which lists oxide components and other variables. The table does not indicate the units of measurement. For oxides, this is most likely Wt%. Further, the reader must guess that "O Total" is the sum of the oxide components.
A separate question on the definition of OH% and H2O%. The author must add information on the determination of the water content (whether these were analytical measurements, or these are calculated data).
What is meant by "N Total"? If this is the total sum of the components of the composition of minerals, taking into account water, then this should be indicated.
Then, columns Fe2, Ti3, Ox and OHy. Again, the reader must guess for himself that we are talking about formula units.
The "Losed Fe" column means "Lost Fe".
There are standards for formatting data in tables. Their examples can be seen already in published articles.
All these calculated data lose their meaning for other phases given in the table. Negative values should be removed from the data array.
Why was the calculation made for 3 formula units of titanium for silicate and other minerals with completely different chemical formulas?
In addition, the work (T. Chernet, L. Pakkanen, 2003) provides a clear algorithm for calculating formula units in the mineral series ilmenite-pseudorutile-leached pseudorutile. T.Chernet, L. Pakkanen (2003) Estimation Of Ferric Iron, Crystal Water And Calculation Of Chemical Formulae For Altered Ilmenite From Electron Microprobe Analyzes, Based On Stoichiometric Criteria. Geological Survey of Finland, Special Paper 36, 23–28. I hope, the author knows this work and will then include it in the list of cited literature.
3. I still have questions about the quality of figure 6. The main peaks on the X-ray diffractograms have a certain slope. There is a feeling that a distortion of the picture has come. In addition, on X-ray diffractograms we see various inscriptions and peak signatures, it is necessary to indicate in the caption to the figure what these designations mean. If among them there are designations of minerals, then they should coincide with the abbreviations already adopted in the article.
4. In addition, I would like to suggest the author to make a diagram of the contents of TiO2 (on one axis) and Fe2O3 + FeO + MgO + MnO (along the other axis). With the amount of analytical data that the author has, you can get interesting pictures of dependencies. And to get the opportunity to trace the trends in the evolution of the compositions of minerals of the ilmetite-
Similar diagrams are presented in the work of (Mucke A.; Chaudhuri J. N. B., 1991), to which the author refers.
I repeat, the data obtained by the author is interesting, but it must be provided in the proper form.
Author Response
Dear Reviewer
The manuscript was revised according to most of the given comments. I hope that I have been able to fulfill most of the requirements in revising the manuscript. The revision was as follows:
Comments and Suggestions for Authors:
- In general, the author did a good job of finalizing the manuscript. The revision of the methodological part deserves special attention. However, there are a few more editorial comments.
- The use of the "&" sign in the designation SiO2&Al2O3 and in other cases according to the text. I would recommend using the spelling of the word "and".
The spelling of the word “and” was used instead of the sign “&”. All of SiO2&Al2O3 were changed to SiO2 and Al2O3:
Page 3, Line 112.
Page 6, Line 238
Page 10, Line 360
Page 11, Line 369
Page 17, Line 476
Page 27, Line 706
- A separate question on the definition of OH% and H2O%. The author must add information on the determination of the water content (whether these were analytical measurements, or these are calculated data).
A paragraph was added in Lines 192-196, Page 5, to explain the nature of H2O:
These constructed Excel softwares includes the calculation of mplecular formula of ilmenite, leached ilmenite, pseudorutile and leached pseudorutile. In pseudorutile and leached pseudorutile Excel softwares, the calculation for both of the number of oxygen anions, number of hydroxyl anions, the content of OH wt%, and hence, the corresponding content of H2O wt% are explained.
- Design of tables.
The top line of the table, which lists oxide components and other variables. The table does not indicate the units of measurement. For oxides, this is most likely Wt%. Further, the reader must guess that "O Total" is the sum of the oxide components.
What is meant by "N Total"? If this is the total sum of the components of the composition of minerals, taking into account water, then this should be indicated.
The top line of each table was modified to include the unit Wt% for analyzed oxides, to explain the columns of calculating the molecular formula, in addition also that the term “losed Fe” was replaced by “lost Fe”.
In fact, the meaning of “OT” and “NT” was explained inside the text in Page 6, Lines 222-223 and Lines 227-228. However, Both of the two terms were explained in the title of the first table:
“Table 1. The microprobe chemical analyses, the corresponding molecular formula and mineral phases of the analyzed spots of the altered brown ilmenite grains; (Figure 1a-1g), the altered brown, yellowish brown&brownish yellow ilmenite grains; (Figure 2a-2n), the altered pale brown, yellow, creamy and white coloured ilmenite grains; (Figure 3a-3i), separated as magnetic at 0.5 ampere value. OT is the original analyzed total oxides sum while NT is the new one after applying the constructed Excel software and calculating the contained H2O wt%”.
- Negative values should be removed from the data array.
- Why was the calculation made for 3 formula units of titanium for silicate and other minerals with completely different chemical formulas?
- In addition, the work (T. Chernet, L. Pakkanen, 2003) provides a clear algorithm for calculating formula units in the mineral series ilmenite-pseudorutile-leached pseudorutile. T.Chernet, L. Pakkanen (2003) Estimation Of Ferric Iron, Crystal Water And Calculation Of Chemical Formulae For Altered Ilmenite From Electron Microprobe Analyzes, Based On Stoichiometric Criteria. Geological Survey of Finland, Special Paper 36, 23–28. I hope, the author knows this work and will then include it in the list of cited literature.
All of negative values were removed from the tables.
The molecular formulas of only pseudorutile and leached pseudorutile which are calculated. They are not calculated for silicate minerals and other mineral phases. Hence, the calculated data as in accordance to the constructed Excel software; for silicate mineral phases were removed. Only the calculated data of the mixed individual phases “TiO2+Fe2O3” which are left to explain the difference between them and that of true psr/lpsr especially the calculated values of “NT”.
Of course, I know the work of Chernet and Pakkanen (2003). In fact, the article of Chernet and Pakkanen (2003), was the base of the calculations for the different molecular formula units in my first article dealing with the strongly magnetic altered ilmenite grains. However, some modifications were introduced in addition also to all the used algorithms were given in detail. Because both of my two articles were sent to the same current journal, the first article was not accepted, T do not repeat some of the cited references in the two articles. Then, the article including the algorithm for calculating formula units which includes the reference of Chernet and Pakkanen (2003).
However, the reference was included in the current article in Page 3, Lines 66-67; [19].
Then all the numbers of the other contained references; from [19] to [40], inside the text and in the references list were changed:
Reference [19] was changed to [20], Page 2, Line70.
Reference [20] was changed to [21], Page 2, Line73.
Reference [21] was changed to [22], Page 2, Line75.
Reference [22] was changed to [23], Page 2, Line75.
Reference [23] was changed to [24], Page 2, Line80; Page 28, Lines 717, 722.
Reference [24] was changed to [25], Page 2, Line81; Page 28, Line 730.
Reference [25] was changed to [26], Page 2, Line82.
Reference [26] was changed to [27], Page 3, Line85.
Reference [27] was changed to [28], Page 3, Line88.
Reference [28] was changed to [29], Page 3, Line91.
Reference [29] was changed to [30], Page 3, Line91.
Reference [30] was changed to [31], Page 3, Line96.
Reference [31] was changed to [32], Page 3, Line125; Page 6, Line 203; Page 28, Line 738.
Reference [32] was changed to [33], Page 3, Line125; Page 4, Line 140.
Reference [33] was changed to [34], Page 21, Line539, 542.
Reference [34] was changed to [35], Page 26, Line684; Page 28, Line 730.
Reference [35] was changed to [36], Page 28, Line717.
Reference [36] was changed to [37], Page 28, Line746.
Reference [37] was changed to [38], Page 30, Line809.
Reference [38] was changed to [39], Page 30, Line813.
Reference [39] was changed to [40], Page 30, Line818.
Reference [40] was changed to [41], Page 30, Line818, 821.
- I still have questions about the quality of figure 6. The main peaks on the X-ray diffractograms have a certain slope. There is a feeling that a distortion of the picture has come. In addition, on X-ray diffractograms we see various inscriptions and peak signatures, it is necessary to indicate in the caption to the figure what these designations mean. If among them there are designations of minerals, then they should coincide with the abbreviations already adopted in the article.
I have tried to refine Figure 6 as closely as I can.
- In addition, I would like to suggest the author to make a diagram of the contents of TiO2 (on one axis) and Fe2O3 + FeO + MgO + MnO (along the other axis). With the amount of analytical data that the author has, you can get interesting pictures of dependencies. And to get the opportunity to trace the trends in the evolution of the compositions of minerals of the ilmetite-Similar diagrams are presented in the work of (Mucke A.; Chaudhuri J. N. B., 1991), to which the author refers.
I apologize for not fulfilling this last request, and I beg you to agree with me on that. In fact, there are many relationships that can be dealt with, but due to the large volume of research, I have moved away from the graphical relationships in the group of three researches related to leucoxene, one of them the current article.
Now, I think that most of the requirements given by the Reviewer in the table of his report are achieved.
With my best regards
M.I.Moustafa
